

SciPost Phys. Lect. Notes 2 (2018)

# Lecture notes on diagrammatic Monte Carlo for the Fröhlich polaron

**Jonas Greitemann and Lode Pollet⋆**

Department of Physics and Arnold Sommerfeld Center for Theoretical Physics,
Ludwig-Maximilians-Universität München, Theresienstr. 37, 80333 Munich, Germany

⋆ lode.pollet@physik.uni-muenchen.de

## Abstract

These notes are intended as a detailed discussion on how to implement the diagrammatic Monte Carlo method for a physical system which is technically simple and where it works extremely well, namely the Fröhlich polaron problem. Sampling schemes for the Green function as well as the self-energy in the bare and skeleton (bold) expansion are disclosed in full detail. We discuss the Monte Carlo updates, possible implementations in terms of common data structures, as well as techniques on how to perform the Fourier transforms for functions with discontinuities. Control over the variety of parameters, especially in the bold scheme, is demonstrated. Sample codes are made available online along with extensive documentation. Towards the end, we discuss various extensions of the method and their applications. After working through these notes, the reader will be well equipped to explore the richness of the diagrammatic Monte Carlo method for quantum many-body systems.

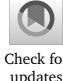

# 1 Introduction

These notes originate from a series of lectures taught at international summer schools intended for researchers interested in numerical methods and strongly correlated systems. They introduce the diagrammatic Monte Carlo (DiagMC) method, a quantum Monte Carlo method for strongly correlated systems in which one, simply put, samples over all Feynman diagrams. Feynman diagrams are versatile and employ a universal language used in high-energy as well as in condensed matter physics. DiagMC is one of the most promising methods still under active development to deal with generic fermionic models in high dimensions. The goal is to give an introduction and flavor of this method.

 Prerequisities for a thorough understanding of this text are familiarity with the basics of quantum mechanics and elementary quantum field theory (notions such as the interaction picture, Wick's theorem, Green function formalism, etc.), statistical mechanics (partition function, solving two-level systems, etc.), and undergraduate computational physics (curve fitting, root solving, interpolation techniques, etc.) including classical Monte Carlo methods (notion of detailed balance, Markov chain Monte Carlo, Metropolis algorithm, etc.).

 Let us summarize the main idea of the method, and how it differs from other quantum Monte Carlo schemes – admittedly, different researchers use the notion of diagrammatic Monte

Carlo in quite different contexts. To this end, we must discuss the type of expansion, the sampling space, and the nature of the sampled series. Newcomers may skip the remainder of this paragraph in a first reading.

The starting point is a very general perturbative expansion of the form,

$$F(y) = \sum_n \sum_{x_1,\ldots x_n} D(x_1,\ldots,x_n;y). \tag{1}$$

We compute a function $F$ depending on external coordinates $y$ (for example, the Green function $G(k,\tau)$ with momentum $k$ and imaginary time $\tau$) which has a perturbative expansion. At every order $n$ there are internal coordinates $x_1,\ldots,x_n$ (these are the internal momenta and imaginary times) which can be discrete or continuous, and are summed or integrated over. The different topologies have different kernels $D$ (cf. Fig. 5 below). The starting point in DiagMC is a weak coupling expansion, *i.e.,* an expansion in the interaction. Let us decompose the Hamiltonian as $H = H_0 + H_1$ where $H_0$ contains all one-body terms (and constitutes hence a quadratic Hamiltonian) and $H_1$ the interactions. Our basis states are the eigenstates of $H_0$. Similar choices are made in (lattice) determinant Monte Carlo simulations for fermions, and in fermionic impurity solvers such CT-INT and CT-AUX (see Ref. [1] for a recent review). By contrast, a "strong-coupling" expansion is used in path-integral Monte Carlo simulations [2], in the worm algorithm [3] and the fermionic impurity solver CT-HYB [1]. In these schemes one perturbs in the kinetic hopping term whereas the solvable system (the potential energy term) is diagonal in the chosen Fock or real-space basis but not quadratic – it corresponds to the atomic limit.

An expansion of the partition function $Z$ at inverse temperature $\beta = 1/T$ and volume $V$ in the sense of a weak-coupling expansion and in the spirit of Eq. 1 reads

$$
\begin{aligned}
Z &= \mathrm{Tr}\, e^{-\beta H} = \mathrm{Tr}\, \mathscr{T}_\tau\, e^{-\beta H_0} \exp\left[ -\int_0^\beta d\tau H_1(\tau) \right] \\
&= \sum_k (-1)^k \int_0^\beta d\tau_1 \ldots \int_{\tau_{k-1}}^\beta d\tau_k \mathrm{Tr}\left[ e^{-\beta H_0} H_1(\tau_k)\ldots H_1(\tau_1) \right],
\end{aligned} \tag{2}
$$

where in the second line we worked out the time-ordering operator $\mathscr{T}_\tau$ of the first line. This expansion leads to nothing but a Taylor expansion in the interaction $H_1$, namely $Z = \sum_k c_k g^k$ with $g$ the coupling strength amplitude of $H_1$ and $c_k$ the coefficients that can be determined by evaluating all the integrals in Eq. 2 order by order, and which remain independent of $g$. Methods such as lattice determinant Monte Carlo and the impurity solvers CT-INT and CT-AUX (but also the Monte Carlo methods referred to as strong-coupling expansions) evaluate physical quantities in thermodynamic equilibrium as

$$\langle Q \rangle = \frac{\mathrm{Tr}\, Q e^{-\beta H}}{Z}, \tag{3}$$

and give it the following statistical meaning: Sample configurations $c$ are obtained, which are distributed according to the partition function $Z$ with respective weights $p_c$, and in which the quantity $Q$ is evaluated. Hence,

$$\langle Q \rangle = \frac{\sum_c Q_c p_c}{\sum_c p_c}. \tag{4}$$

The unbiased estimator for the expectation value of the quantity $Q$ is then to sum up $Q_c$ over all independent configurations and divide by the number of independent measurements. The normalization through the partition function is here manifest. As long as the system volume $V$ and its inverse temperature $\beta$ are finite, the Eq. 2 is an expansion in an entire function and

hence always convergent (with the finiteness of the system we explicitly exclude all possible UV divergences that may still arise as e.g. in Sec. 8.2). The finiteness of the system ensures that no true spontaneous symmetry breaking can occur, which is at the heart of such methods as finite size scaling.

When physicists use the term DiagMC in the sense of the expression "sampling over all Feynman diagrams" it implies a number of differences compared to the previous paragraph: The thermodynamic limit is taken from the start, the partition function is usually not used for normalization (instead, the lowest order diagram is often chosen (see below)), nor does the sampling necessarily take place in the space of the partition function diagrams: The method (usually) relies on the cancellation of disconnected diagrams when computing correlation functions as can be found in standard textbooks [4–8]. This can equivalently be considered an expansion of the free energy $F \sim \log Z$.

These differences allow us to sketch some of the key properties of Feynman diagrams, which can be considered its advantages: All diagrams are topologically distinct and the magnitude of the prefactor is always 1 [6]. The language of Feynman diagrams is universal in all fields of physics. Feynman diagrams factorize over internal building blocks, such as particle propagators (single particle Green functions), interactions, and vertices. Consequently, the diagram weight also factorizes, which is a prerequisite for successfully developing a Markov chain Monte Carlo method. Analytical treatments of low orders or limiting cases can be built in analytically. In DiagMC one does not attempt to write down all diagrams explicitly (since the number of diagrams grows factorially with expansion order, this is only possible for the lowest expansion orders anyway) but one instead develops algorithmic rules that allow one to sample over all diagrams. This implies changing the internal integration variables, but also the topology and the expansion order. Non-perturbative features are accessible via skeleton series [9] and (partial) resummations of a certain class of diagrams. This takes us away from the bare expansion, and we will also see how this works for the Fröhlich polaron. In fact, any analytical treatment known from the literature can be built in. Ideally, the Monte Carlo sampling should only deal with featureless functions originating from high-dimensional integrals whereas any intricacy related to the field theory is dealt with analytically a priori.

The aforementioned differences bring us at the same time to the first main difficulty in the development of the DiagMC method, which is the series convergence: It is usually unknown whether a series converges or not. The series is guaranteed to diverge at a phase transition, but it may happen sooner. In fact, most series in physics are asymptotic, which can be established rigorously in a number of cases. A well known argument, first formulated in the context of quantum electrodynamics, is Dyson's collapse argument [10]: When rotating the electric charge from $e$ to $ie$ in the complex plane around the origin, one sees that the system is unstable to collapse (the potential energy scales quadratically with the number of particles, which is faster than the kinetic energy), rendering the convergence radius zero. The same holds for any interacting bosonic field theory: No matter how small in magnitude the attraction in the potential energy is, it beats the kinetic energy for large enough particle numbers, leading to a collapse. The asymptotic nature of the series can sometimes be dealt with using resummation methods [8], but, in general, the issue of a non-convergent series is an open problem and in our view the most difficult one that DiagMC faces.

The second main difficulty in the development of DiagMC is the sign problem. Sign alternations are often inherent (and necessary) to the issue of convergence – without sign alternations the factorial growth in the number of diagrams could never lead to a meaningful result for an asymptotic series. Nor is the sign extensive in the system volume, as in path integral Monte Carlo simulations, which would prohibits us from finding the full solution [11]. Nevertheless, the sign problem puts in practice a limit on the expansion orders that can be reached. DiagMC features hence a tacit assumption that the sign problem is sufficiently weak such that

sufficiently high expansion orders can be reached in order to extrapolate in a reliable way to infinite expansion orders (often in combination with a resummation scheme that is powerful enough). Unfortunately, this assumption can only a posteriori be checked.

The third difficulty is dealing with multi-dimensional objects such as a multi-legged vertex in the Bethe-Salpeter equation. Despite active research in the fields of self-adaptive grids and concise data storage formats, this is equally an unsolved problem. However, only in cases that an explicit expression for the whole object (or a high-dimensional subpart) is required (such as in self-consistency schemes) can this be considered a problem; otherwise one can just sample over such an object without ever evaluating it in full.

In these notes we consider a model where these three problems do not occur: the Fröhlich polaron model is sign-positive in the imaginary time formalism and the Green function convergent for all finite values of the imaginary time. Due to the rotational symmetry of free space can the Green function be stored as a two-dimensional object, which is easy to histogram and manipulate. There are other simplifying factors, which are related to the absence of vacuum polarization diagrams, or, equivalently, the observation that Feynman diagrams for polaron (and impurity) problems can be mapped onto path integrals (cf. the structure of a backbone line in Fig. 5 below). Indeed, the analytical properties of mesoscopic systems such as polarons and impurity systems appear to be much simpler than those of true many-body problems. Furthermore, for almost all problems of this type very accurate variational approaches (and wavefunctions) are known. The Fröhlich polaron is hence ideal to get acquainted with the DiagMC method. Not suprisingly, it was also the first model to which the method was applied 20 years ago [12, 13].

This text is structured as follows. After discussing perturbative expansions with continuous variables in Sec. 2, the main body of this text deals with the Fröhlich polaron problem, whose Green function is obtained from a bare expansion in Sec. 3, the self-energy from a bare expansion in Sec. 4 and from a bold expansion in Sec. 5. The source codes are made publicly available as discussed in Sec. 6. In Sec. 8 some related physical systems (of the polaron or impurity type) are listed where the acquired techniques can (and have been) applied without going into detail about the physics. For completeness, we mention that the method has also been successfully applied to a number of problems that cannot be considered of the polaron or impurity-type leading to deeper insight in notoriously hard problems. We mention resonant fermions [14–16], frustrated magnetism [17–19], and physics found in the Hubbard model [20–23], among others.

## 2 Continuous-time Monte Carlo

It is quite common to have discrete as well as continuous variables in quantum field theory. In this first section we explain, by means of the celebrated two-level system, how continuous variables and variable expansion orders can be dealt with in a Monte Carlo sampling. We employ the path integral representation here.

### 2.1 Model

Consider a two-level system with Hamiltonian,

$$H = H_0 + H_1 = h\sigma_z + \Gamma\sigma_x \qquad h, \Gamma > 0, \tag{5}$$

where $\sigma_x = \begin{pmatrix} 0 & 1 \\ 1 & 0 \end{pmatrix}$ and $\sigma_z = \begin{pmatrix} 1 & 0 \\ 0 & -1 \end{pmatrix}$ are the usual Pauli matrices in the $z$-basis with basis states $|\uparrow\rangle = \begin{pmatrix} 1 \\ 0 \end{pmatrix}$ and $|\downarrow\rangle = \begin{pmatrix} 0 \\ 1 \end{pmatrix}$. The $h$-field tries to orient the spin along the $-z$-axis which is

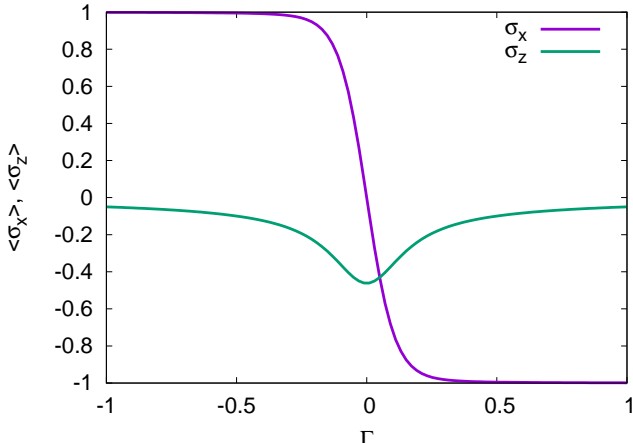

Figure 1: Magnetization along the $x$ and the $z$ axis for $\beta = 10$, $h = 0.05$ and variable $\Gamma$.

countered by the $\Gamma$-field which tries to orient the spin along the $-x$-axis. This system can be solved exactly, with the solutions (shown in Fig, 1)

$$
\begin{aligned}
\langle \sigma_x \rangle &= \frac{-\Gamma}{\sqrt{\Gamma^2 + h^2}} \tanh\left( \beta \sqrt{\Gamma^2 + h^2} \right), \\
\langle \sigma_z \rangle &= \frac{-h}{\sqrt{\Gamma^2 + h^2}} \tanh\left( \beta \sqrt{\Gamma^2 + h^2} \right),
\end{aligned}
\tag{6}
$$

$$
\tag{7}
$$

which makes this system a good model to get acquainted with continuous-time Monte Carlo. From the symmetry of the Hamiltonian we see that we can swap $x \leftrightarrow z$ if we also swap $\Gamma \leftrightarrow h$.

## 2.2 Perturbative expansion

Starting from the partition function

$$
Z = \mathrm{Tr} e^{-\beta H} = \sum_{|\alpha\rangle = |\uparrow\rangle, |\downarrow\rangle} \left\langle \alpha \left| e^{-\beta(\Gamma \sigma_x + h \sigma_z)} \right| \alpha \right\rangle,
\tag{8}
$$

we notice that the $\sigma_z$ operator is diagonal in this basis. In order to prepare for a perturbative expansion in the $\Gamma \sigma_x$ term, we introduce the Heisenberg operators

$$
\sigma_x(\tau) = e^{h \sigma_z \tau} \sigma_x e^{-h \sigma_z \tau},
\tag{9}
$$

and rewrite the partition function as

$$
Z = \sum_{|\alpha\rangle = |\uparrow\rangle, |\downarrow\rangle} \sum_{n=0}^{\infty} \frac{(-\Gamma)^n}{n!} \int_0^\beta d\tau_1 \dots \int_0^\beta d\tau_n \left\langle \alpha \left| e^{-\beta h \sigma_z} \mathcal{T}_\tau [\sigma_x(\tau_1) \dots \sigma_x(\tau_n)] \right| \alpha \right\rangle.
\tag{10}
$$

This is an explicit formulation of Eq. 2, $Z = \mathrm{Tr}\, \mathcal{T}_\tau \exp(-\beta H_0) \exp(-\int_0^\beta H_1(\tau) d\tau)$.

To lowest order ($n = 0$) there are just 2 contributions, $Z_0 = \exp(-\beta h) + \exp(\beta h)$. Graphically, this can be depicted as a continuous worldline from $\tau = 0$ to $\tau = \beta$ (see panel (a) in Fig. 2). We use a full line for spin-up and a dashed line for spin-down. Note that worldlines are continuous and periodic in $\beta$ because of the cyclical properties of the trace. For this reason, there are no non-zero contributions for $n = 1$, nor for any odd value of $n$. This means that the

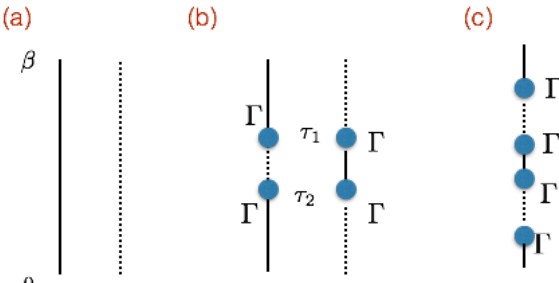

Figure 2: (a) The $0^{\text{th}}$ order contributions to the partition function of the two-level system. Spin-up worldlines are represented by a full line, spin-down worldlines by a dashed line. (b) The second order contributions, and (c) the general structure. Worldlines are periodic over the imaginary time period $\beta$.

term $(-\Gamma)^n$ in Eq. 10 is always positive and we do not have to worry about a sign problem. In second order ($n = 2$) (see panel (b) in Fig. 2) we have

$$Z_2 = \Gamma^2 \sum_{\substack{|\alpha\rangle=|\uparrow\rangle,|\downarrow\rangle \\ |\alpha_1\rangle=|\uparrow\rangle,|\downarrow\rangle}} \int_0^\beta \mathrm{d}\tau_1 \int_0^{\tau_1} \mathrm{d}\tau_2 \left\langle \alpha \left| e^{-(\beta-\tau_1+\tau_2)h\sigma_z}\sigma_x \right| \alpha_1 \right\rangle \left\langle \alpha_1 \left| e^{-(\tau_1-\tau_2)h\sigma_z}\sigma_x \right| \alpha \right\rangle. \quad (11)$$

Note that there is no factor of $1/2$ because it cancelled with the number of equivalent contributions from the time ordering operator and the corresponding changes in the time integration boundaries [6]. Although the higher order terms can be written in the same fashion, the integrals quickly become too complicated to evaluate explicitly. We therefore switch to a stochastic approach, for which it is easiest to think in terms of a graphical depiction, as shown in panel (c) of Fig. 2. To a vertex we attribute a factor $\Gamma$, and to each segment of length $\tau$ (measured taking the periodic boundary conditions in $\beta$ into account) we attribute a weight $\exp(\pm\tau h)$ with the sign depending on the spin state.

Analyzing the limiting cases, we expect to find, with almost equal probability, worldlines that are dominated by one of the spin states with few vertices at high temperatures. At low temperatures, we expect a dashed line with few kinks for $\Gamma \ll h$, whereas for $h \ll \Gamma$ the spin wants to orient along the $-x$-direction, which graphically translates into having many vertices, and for which our chosen basis along the $z$-direction is a poor choice. Our main task when designing a Monte Carlo scheme is hence to reach high expansion orders at low temperature with good efficiency.

## 2.3 Monte Carlo updates

There exist many equivalent ways to sample this system. The choice we make here resembles the updates later used in the Fröhlich polaron code, with similar design criteria. A minimal ergodic set of updates consists of the pair INSERT/REMOVE. If the INSERT update is chosen, we attempt to insert a new pair of vertices as shown in Fig. 3. We therefore select a random time $\tau_1$ chosen uniformly over $\tau_1 \in [0,\beta[$. Looking in the direction of positive imaginary times, we determine the time interval $\Delta$ counted from $\tau_1$ over which the spin occupation does not change. The second vertex is placed at a time chosen uniformly over the interval $\Delta$. For the reverse update, the pair to be removed consists of randomly selecting a vertex and taking the subsequent one in the direction of positive time. The weight of the diagram segment between $\tau_1$ and $\tau_2$ in the old (i.e., before the INSERT update) configuration $X$ is $W(X) = e^{-(\tau_2-\tau_1)hn_0}$, with $n_0$ the spin occupation at time $\tau_1$ in the old configuration. The weight of the corresponding segment

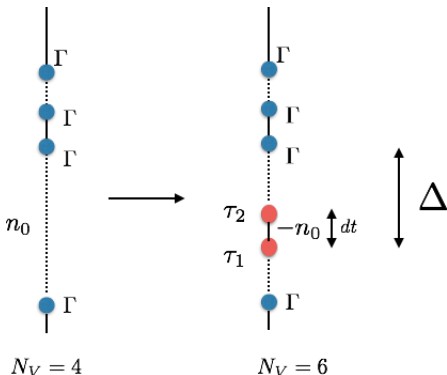

Figure 3: Illustration of the INSERT/REMOVE pair of updates for the two-level system.

in the new configuration $Y$ is $W(Y) = \Gamma^2 e^{(\tau_2-\tau_1)hn_0} d\tau_1 d\tau_2$. With equal probabilities of selecting the INSERT and REMOVE updates, the probability factors are $P(X \to Y) = \frac{1}{\beta\Delta} d\tau_1 d\tau_2$ and $P(Y \to X) = \frac{1}{N_V+2}$ with $N_V$ the number of vertices present in the old configuration. The update INSERT is accepted according to the Metropolis algorithm with probability $\min(1, r)$ where the acceptance factor $r$ is given by $r = \frac{W(Y)P(Y \to X)}{W(X)P(X \to Y)}$. For the REMOVE update the acceptance factor is $1/r$. The differentials $d\tau_1$ and $d\tau_2$ enter the formulas for $W(Y)$ and $P(X \to Y)$ as a consequence of working with a continuous variable $\tau$ but they drop out in the acceptance factor $r$.

## 2.4 Estimators

The observables of interest are the expectation value of the spin magnetization along the $x$- and the $z$-axis. There are two ways to measure the magnetization along the $z$-axis (up to an irrelevant minus sign). The first one consists of evaluating the magnetization at a fixed time $\tau = 0$, which is an integer number. The second one evaluates the integral $\frac{1}{\beta}\int_0^\beta \sigma_z(\tau)d\tau$, which is a floating point number. Perhaps the reader thinks that the second way is far superior because it contains information from all times, but we will see that this is not true: the second way of measuring has only slightly lower error bars for the same runtime, whereas it is a considerably more expensive operation to perform, scaling linearly in the number of interaction vertices (even if done "on the fly" after every update). The magnetization along the $x$-direction can be measured as $\langle\sigma_x\rangle = -\frac{\langle N_V\rangle}{\beta\Gamma}$ as can be seen from Eq. 10. It is equally straightforward to obtain estimators for quantities such as $\langle\sigma_x(\tau)\sigma(0)\rangle$, but we will not discuss this further.

## 2.5 Results

Let us start at low temperature with a strong magnetic field in the $x$-direction. We take as parameters $\beta = 10, \Gamma = 0.4, h = 0.05$. After an initial thermalization phase of one million updates, we perform 10,000,000 updates, measuring after each one. After just a few seconds we see that we reproduce the exact result with error bars between 0.0001 and 0.001. The integrated autocorrelation times are about 3. We spend about 4% of the time in the zeroth order diagram, and close to 40% of the time in fourth order, although the code has occasionally gone to 16th order. So we sampled over quite a large Hilbert space and the code performed very well. There is no reason to optimize further.

At low temperature and strong magnetic field in the $z$-direction ($\beta = 10, \Gamma = 0.05, h = 0.4$ and same runtime parameters as before) the autocorrelation times are also about 3. The error bars on $\langle\sigma_z\rangle$ are typically an order of magnitude smaller than in the case $\Gamma \gg h$, which is

explained by the fact that our basis is better suited. The error bars on $\langle \sigma_x \rangle$ are only slightly larger than before. In other words, the code is still behaving as expected.

At high temperature ( $\beta = 0.1, \Gamma = 0.2, h = 0.2$ and same runtime parameters as before) the magnetization along either direction is about $-0.08$ and hence very weak. Perhaps surprisingly, we see that the error bar on $\langle \sigma_z \rangle$ is of the order of 0.04, which is 50 times larger than the error bar on $\langle \sigma_x \rangle$ and one to two orders of magnitude larger than what we had at low temperatures – whereas low temperatures should be much more difficult to simulate. This is also reflected in the integrated autocorrelation times, which are about 8,000 (it could well be worse because it is not clear if the code has converged) for the magnetization along the $z$-axis and only $\sim 1$ along the $x$-axis. Physically, the system has rotational symmetry in spin space, but this is clearly not respected in our updating procedure. As expected, the code spends 99.98% of the time in the zeroth order diagram and the acceptance ratio for our INSERT-REMOVE updates is 0.02%. What could be the reason for such bad autocorrelation times in an essentially non-interesting regime? The world-lines are 99.98% of the time straight world-lines but the up and down orientations are almost equally probable because of the high temperature. Our current update scheme only allows one to change the orientation of the magnetization via the insertion of kinks, which is highly inefficient at high temperature. To cure this problem, we add another update SPIN-FLIP which, for simplicity, is only allowed in the zero-vertex sector and which attempts to swap between the up and down orientations of the spin. Adding this update cures the problem. It is good practice to keep the code as simple (and local) as possible, and to optimize or write extra updates only in case problems pop up.

With this we close the discussion on sampling continuous variables and different expansion orders and proceed to the main part.

# 3 Fröhlich polaron: bare expansion for the Green function

The Fröhlich polaron problem describes the interaction between an itinerant electron and longitudinal, optical phonons. Historically, it was the first problem to which diagrammatic Monte Carlo was applied [12, 13, 24] for which it could provide definite answers regarding the polaron spectrum and arbitrarily precise polaron energies for any coupling strength. The Hamiltonian for a system in a volume $V$ is given by

$$
\begin{aligned}
H &= H_{\mathrm{el}} + H_{\mathrm{ph}} + H_{\mathrm{el-ph}}, \\
H_{\mathrm{el}} &= \sum_{\mathbf{k}} \frac{(\hbar \mathbf{k})^2}{2m} a_{\mathbf{k}}^{\dagger} a_{\mathbf{k}}, \\
H_{\mathrm{ph}} &= \sum_{\mathbf{q}} \hbar \omega_{\mathbf{q}} b_{\mathbf{q}}^{\dagger} b_{\mathbf{q}} = \hbar \omega_{\mathrm{ph}} \sum_{\mathbf{q}} b_{\mathbf{q}}^{\dagger} b_{\mathbf{q}}, \\
H_{\mathrm{el-ph}} &= \sum_{\mathbf{k,q}} V(\mathbf{q})(b_{\mathbf{q}}^{\dagger} - b_{\mathbf{q}}) a_{\mathbf{k-q}}^{\dagger} a_{\mathbf{k}}, \\
V(\mathbf{q}) &= i \frac{\hbar \omega_{\mathrm{ph}}}{q} \left( \frac{4\pi\alpha}{V} \right)^{1/2} \left( \frac{\hbar}{2m\omega_{\mathrm{ph}}} \right)^{1/4}.
\end{aligned}
\tag{12}
$$

The operators $a_{\mathbf{k}}$ and $b_{\mathbf{q}}$ are annihilation operators for electrons of mass $m$ with momentum $\mathbf{k}$ and phonons with momentum $\mathbf{q}$, respectively. The phonon frequency $\omega_{\mathbf{q}} = \omega_{\mathrm{ph}}$ can be taken momentum-independent for optical, longitudinal phonons. The dimensionless coupling constant is $\alpha$. Typical values for $\alpha$ vary from 0.023 for InSb over 0.29 for CdTe to 1.84 for AgCl (and are thus rather weak) [25]. We will work in units $\hbar = m = \omega_{\mathrm{ph}} = 1$ and take the continuum limit $\frac{1}{V} \sum_{\mathbf{q}} \to \int \frac{d^3\mathbf{q}}{(2\pi)^3}$.

It is not the purpose of these notes to give an overview of the physics of the Fröhlich polaron, whose thermodynamics is now well understood (but questions remain for transport). We refer to the lecture notes by J. Devreese [25] for a pedagogical introduction. The basic competition in the model is between the electron kinetic energy trying to delocalize the particle and the phonons trying to localize it. The system can lower its energy by dressing the electron with phonons, resulting in the formation of a polaron. Its residue can be very low and the effective mass very high, but the polaron is never fully localized or fully self-trapped; there is hence no transition in this model.

For historical importance and to illustrate the connection with path integrals, let us remark that the Hamiltonian is quadratic in the phonon propagators, which can hence be integrated out. This results in a retarded one-particle propagator for the electron,

$$\langle 0,0|0,\beta\rangle = \int D\mathbf{r}(\tau)\exp\left[-\frac{1}{2}\int_0^\beta \dot{\mathbf{r}}(\tau)^2 d\tau + \frac{\alpha}{2^{3/2}}\int_0^\beta \int_0^\beta \frac{e^{-|\tau-\sigma|}}{|\mathbf{r}(\tau)-\mathbf{r}(\sigma)|}d\tau d\sigma\right], \qquad (13)$$

where $|\mathbf{r},\tau\rangle$ is in the basis of position and imaginary time. Thus, Eq. 13 conveys the intuitive idea of obtaining the probability amplitude for an electron to return to its initial position after an imaginary time evolution up to inverse temperature $\beta$ by integrating over all possible trajectories ('paths') through imaginary time.

This path integral expression served as the basis of Feynman's variational ansatz [26] which is remarkably accurate for the polaron energy for all coupling strengths. This path integral is, because of the retarded self-interaction, not as easy to simulate as the two-level system of the previous section, and will hence not be used for actual computations.

The structure of this section is as follows: We start with reviewing the necessary field-theoretical formulas to study quasi-particle properties, followed by the description of the algorithm used to simulate the polaronic Green function using a bare expansion. Next, we show some results that can be obtained with this code. In the following section the self-energy is computed using the bare expansion, with special emphasis on Fourier transforms and an illustration for the first-order diagram. Finally, the bold expansion of the self-energy is introduced, again splitting the discussion between the first-order diagram and higher order ones.

## 3.1 Digest of many-body theory

The central object of our analysis is the full single-particle Green function, which is related to the bare Green function $G_0$ and the self-energy $\Sigma$ via the Dyson equation as

$$G^{-1}(\mathbf{k},\omega_n) = G_0^{-1}(\mathbf{k},\omega_n) - \Sigma(\mathbf{k},\omega_n). \qquad (14)$$

For the polaron problem, we will work at zero temperature. To avoid instabilities due to poles, it is more convenient to work in imaginary time than with Matsubara frequencies $\omega_n$ in the sampling. For impurity problems, the bare Green function is just

$$G_0(\mathbf{k},\tau) = -\theta(\tau)e^{-(\epsilon_k-\mu)\tau}, \qquad (15)$$

with $\theta(\cdot)$ the Heaviside function, $\epsilon_k = \frac{k^2}{2m}$ the dispersion, and $\mu$ an energy shift which is used as a tuning parameter (see below). In Matsubara representation the bare Green function takes the form

$$G_0(\mathbf{k},\omega_n) = \frac{1}{i\omega_n - (\epsilon_k-\mu)}. \qquad (16)$$

The full Green function will have a pole at $i\omega_n = E_k - \mu$ where $E_k$ is the self-consistent solution to

$$E_k = \epsilon_k + \Sigma(\mathbf{k},E_k-\mu) = \epsilon_k + \int_0^\infty \Sigma(\mathbf{k},\tau)e^{(E_k-\mu)\tau}d\tau, \qquad (17)$$

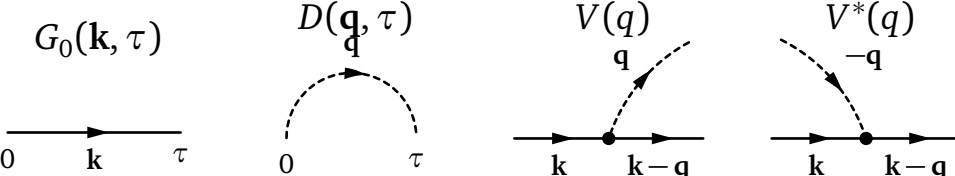

Figure 4: Bare diagrammatic elements for the Fröhlich polaron. From left to right, shown are the bare Green function, the phonon propagator, and the 2 vertices where a phonon is emitted (absorbed) causing a shift of the electron momentum.

given that the imaginary part of $\Sigma(\mathbf{k}, E_k - \mu)$ vanishes. We may then expand the self-energy around the pole position,

$$\Sigma(\mathbf{k}, \omega_n) = \Sigma(\mathbf{k}, E_k - \mu) + \partial_{i\omega_n}\Sigma(\mathbf{k}, E_k - \mu)(i\omega_n - E_k + \mu) + \mathcal{O}(|i\omega_n - E_k + \mu|^2), \quad (18)$$

allowing us to rewrite the full Green function approximately as

$$G(\mathbf{k}, \omega_n) = \frac{1}{i\omega_n - (\epsilon_k - \mu) - \Sigma(\mathbf{k}, \omega_n)} \approx \frac{Z_k}{i\omega_n - (E_k - \mu)} \quad (19)$$

with the quasi-particle residue

$$Z_k = \frac{1}{1 - \partial_{i\omega_n}\Sigma(\mathbf{k}, E_k - \mu)} = \left(1 - \int_0^\infty \tau\Sigma(\mathbf{k}, \tau)e^{(E_k - \mu)\tau}d\tau\right)^{-1}. \quad (20)$$

The approximation Eq. 19 holds as long as the quasi-particle pole is sufficiently far away from the dissipative continuum, the separation to which we call $\Delta$. Transforming back to imaginary time, the quasi-particle energy $E_k$ and residue $Z_k$ (which is the modulus squared of the overlap between the quasi-particle state and the free electron state) can be extracted from the large $\tau$ behavior of the full Green function under the same assumptions,

$$G(\mathbf{k}, \tau) \to -\theta(\tau)Z_k e^{-(E_k - \mu)\tau} \qquad \text{for} \quad \tau \gg \Delta^{-1}. \quad (21)$$

We will solve the problem of obtaining $\Sigma(\mathbf{k}, \tau)$ for fixed $\mu$ by diagrammatic Monte Carlo and are left with the task of finding $E_k$ such that Eq. 17 is satisfied. This can be done by a root-solving algorithm in combination with one-dimensional integration. When $E_k$ is found self-consistently, Eq. 20 determines the corresponding residue. The dispersion of the quasi-particle is given by analyzing $E(k)$ as a function of $k$.

## 3.2 Algorithm

The simplest way to solve the Fröhlich polaron problem is by considering the bare expansion of the full green function by using the expansion elements shown in Fig. 4. This was also presented in the original solution by Prokof'ev and Svistunov [12]. Wick's theorem tells us that there can be no unpaired phonon creation and annihilation operators, i.e., all phonon operators pair into 'arcs', the number of vertices is always even and $V(\mathbf{q})$ in Eq. 12 only enters as a product with its complex conjugate. Graphically, the expansion is illustrated in Fig. 5. We can label the expansion order by counting the number of phonon propagators. In order $n$ there are $n$ phonon propagators, $2n$ vertices and $2n + 1$ impurity Green functions. Our task consists of sampling over all possible diagrams for the Green function $G(\mathbf{k}, \tau)$, i.e., sample over all possible expansion orders $n$, all allowed topologies, and integrate over all internal momenta $\mathbf{q}_i, i = 1, \ldots, n$, and vertex times $\tau_j, j = 1, \ldots, 2n$.

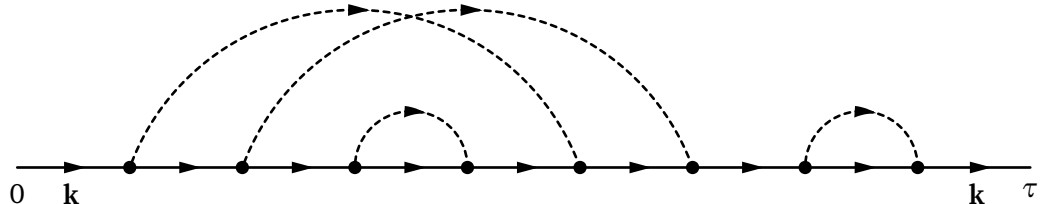

Figure 5: Bare expansion of the full Green function $G(\mathbf{k}, \tau)$ for the Fröhlich polaron. A non-interacting Green function is denoted by a thin full 'backbone'-line, a phonon propagator by a dashed arc and a vertex by a dot. The expansion order $n$ is given by counting the number of arcs (here, $n = 4$). Only connected diagrams are considered. This diagram is a typical example of a 'configuration' whose weight is the product of the weights of the composing Green functions, vertices and phonon propagators.

Every Feynman diagram is a valid Monte Carlo configuration, with a weight that factorizes into the product of the individual electron propagators $G_0(\mathbf{k}, \tau)$, phonon propagators

$$D(\mathbf{q}, \tau) = \exp(-\omega_{\text{ph}}\tau) \tag{22}$$

and vertices. It is convenient to absorb the $1/q$ vertex dependence into the phonon propagators $\tilde{D}(\mathbf{q}, \tau) = D(\mathbf{q}, \tau)/q^2$ and constants into the coupling constant $\tilde{\alpha}^2 = 2\pi\alpha\sqrt{2}$, see Eq. 12 and our choice of units.

As an example, the full expression for the weight $W_c^{(2)}$ of the second order diagram with crossing phonon lines (one of the three possible topologies in second order) reads

$$W_c^{(2)}(\mathbf{p}, \tau) = - \tag{23}$$

$$= -\tilde{\alpha}^4 \int_0^\tau d\tau_1 \int_{\tau_1}^\tau d\tau_2 \int_{\tau_2}^\tau d\tau_3 \int_{\tau_3}^\tau d\tau_4 \int \frac{d^3\mathbf{p}_1}{(2\pi)^3} \int \frac{d^3\mathbf{p}_2}{(2\pi)^3}$$

$$\times G_0(\mathbf{p}, \tau_1) G_0(\mathbf{p}_1, \tau_2 - \tau_1) G_0(\mathbf{p}_2, \tau_3 - \tau_2) G_0(\mathbf{p}_3, \tau_4 - \tau_3) G_0(\mathbf{p}, \tau - \tau_4)$$

$$\times \tilde{D}(\mathbf{q}_1, \tau_3 - \tau_1) \tilde{D}(\mathbf{q}_2, \tau_4 - \tau_2). \tag{24}$$

Here $\mathbf{p}$ and $\tau$ are the external momentum and time, respectively. The independent momenta are chosen as $\mathbf{p}_1$ and $\mathbf{p}_2$, whereas $\mathbf{q}_1 = \mathbf{p} - \mathbf{p}_1$, $\mathbf{q}_2 = \mathbf{p}_1 - \mathbf{p}_2$, and $\mathbf{p}_3 = \mathbf{p}_2 + \mathbf{p} - \mathbf{p}_1 = \mathbf{p} - \mathbf{q}_2$ follow through momentum conservation. The factorization of the weight into bare Green functions and phonon propagators is now manifest. The extension of such explicit analytical formulas to higher order is however cumbersome in comparison to drawing the Feynman diagrams.

We proceed therefore to how the diagrammatic Monte Carlo sampling can be performed. The updating scheme discussed below differs from the one introduced originally by Prokof'ev and Svistunov. Using the freedom which every designer of a Monte Carlo procedure has, we seek the simplest set of updates that is ergodic and remains as local as possible. By locality we mean that the number of changes to the current configuration is minimal and only involves one diagrammatic element plus its adjacent elements.

*External variables* – Because of the spherical symmetry of the Hamiltonian, we can choose the orientation of the external momentum $\mathbf{k}$ to be along the $x$-axis as $(k, 0, 0)$, in which case the full Green function is a two-dimensional object in $(k, \tau)$ space. We can predefine a set of external momenta $k_j$ for which we compute $G$. The simplest choice is a uniform grid, $k_j = j\Delta_k$ with $\Delta_k = k_{\text{max}}/N_k$ where $k_{\text{max}}$ the momentum cutoff and $N_k$ the number of momentum points.

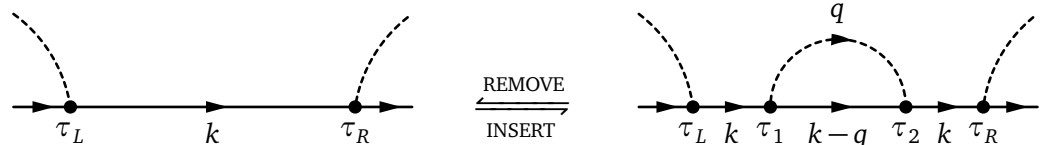

Figure 6: Illustration of the INSERT-REMOVE pair of updates.

Other choices for the grid are possible and perhaps even better because for low momenta we expect a dispersion akin to $k^2/(2m^*)$ with $m^*$ the effective mass of the polaron, which suggests that a quadratic grid might be better. We will not go into further detail. The same procedure can be applied for the external time $\tau$. Let us follow here another approach and consider $\tau$ to be a continuous coordinate between 0 and $\tau_{\max}$, which we bin into a uniform grid of bin length $\Delta_\tau$ at the expense of a small systematic discretization error $\mathcal{O}(\Delta_\tau^2)$ if we use the discretized $\tau_j = (j + 1/2)\Delta\tau$. A logarithmic grid might be a better choice given Eq. 21 (from experience we know that it does not really matter at this stage).

*Normalization* – We choose the zeroth order diagram for normalization, which is just the bare propagator $G_0$. Because of the updates CHANGE-P and CHANGE-TAU (see below) the total normalization integral is $\mathcal{N} = -\sum_{p_j} \int_0^{\tau_{\max}} G_0(p_j, \tau)\mathrm{d}\tau$. All quantities in the Monte Carlo sampling can be normalized by multiplying with $\mathcal{N}/C_0$ where $C_0$ counts how often we are in the zeroth order diagram. This can be seen as follows: The estimator for the normalization diagram is

$$\langle\delta_{\mathrm{norm}}\rangle_{\mathrm{MC}} \propto -\sum_{\{p_j\}} \int_0^{\tau_{\max}} G_0(p_j, \tau)\mathrm{d}\tau. \tag{25}$$

The estimator for the full Green function is

$$\left\langle\mathscr{E}_G(p_j, \tau_i)\right\rangle_{\mathrm{MC}} \propto \delta_{\tau\in\mathrm{bin_i}}\delta_{p_j,p_k}, \tag{26}$$

where we used that $G(p_j, \tau_i) = \sum_{p_k} \int_0^{\tau_{\max}} G(p_k, \tau)\delta(\tau - \tau_i)\delta_{p_j,p_k}\mathrm{d}\tau$ and all $\tau$ residing in the bin $i$ are taken together in entry $\tau_i$ (it is possible to improve on this by computing the ratio $G(p_j, \tau_i)/G(p_j, \tau)$, although there is seldomly a need for that). The proportionality constant drops out when normalizing

$$G(p_j, \tau_i) = -\frac{\left\langle\mathscr{E}_G(p_j, \tau_i)\right\rangle_{\mathrm{MC}}}{\langle\delta_{\mathrm{norm}}\rangle_{\mathrm{MC}}}\mathcal{N}/\Delta\tau_i, \tag{27}$$

with $\Delta_i$ the volume of the time-bin $i$. The same normalization can be applied to other quantities of interest such as the bare Green function and the first-order Green function.

*CHANGE-P* – This update is only allowed if the expansion order is 0. In this update, which is its own reverse, we uniformly select a new $p_j$ from the set of allowed external momenta and accept it according to the Metropolis algorithm as $\min[1, r]$ with acceptance factor $r = G_0(p_j^{\mathrm{new}}, \tau)/G_0(p_j^{\mathrm{old}}, \tau)$. We can also opt to keep the external momentum fixed in a single run.

*CHANGE-TAU* – This update is only allowed if the expansion order is 0. In this update, which is its own reverse, we select a new external time $\tau$ using an exponential distribution. If the dispersion is $\xi_p = \epsilon_p - \mu$ with $p$ the external momentum and $u$ a random number uniformly chosen between $[0, 1[$, then we construct $\tau = -\ln u/\mathrm{abs}(\xi_p)$ and accept it as the new external time of the diagram if $\tau < \tau_{\max}$.

*INSERT* – This update attempts to increase the number of phonon propagators by one (its reverse is REMOVE, see below and Fig. 6) and is constructed as follows: Select a ran-

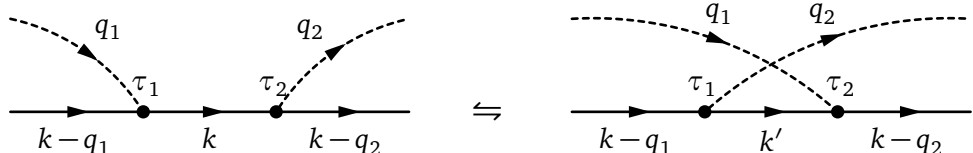

Figure 7: Illustration of the SWAP update. The phonon propagators carrying momentum $q_1$ and $q_2$ are understood to connect back to the fermion line at vertices at times $\tau_A$ and $\tau_B$, respectively. $\tau_A$ and $\tau_B$ may each be either before or after $\tau_1$ and $\tau_2$. After the update, the central fermion propagator carries a momentum $k' = k - q_1 - q_2$.

dom electron propagator and identify its left and right endpoints. Let us call this propagator $G_0(\mathbf{k}, \tau_R - \tau_L)$. Select with uniform probability a time $\tau_1 \in ]\tau_L, \tau_R[$, which serves as the time of the left vertex of the new phonon propagator. The time $\tau_2$ is obtained as $\tau_2 = \tau_1 - \ln u / \omega_{ph}$ with $u \in ]0,1[$ chosen uniformly. If $\tau_2 > \tau_R$ the update is rejected. The three components of the momentum $\mathbf{q}$ are obtained from the Box-Müller algorithm as a gaussian random number with mean 0 and variance $m/(\tau_2 - \tau_1)$. The acceptance factor is given by $r = W(Y)P(Y \to X)/W(X)P(X \to Y)$ where $X$ stands for the old configuration and $Y$ for the new one, $W(\cdot)$ for their respective weights, and $P(\cdot)$ for their a priori transition probabilities. They are given by

$$
\begin{aligned}
W(X) &= -G_0(\mathbf{k}, \tau_R - \tau_L), \\
W(Y) &= -\tilde{\alpha}^2 G_0(\mathbf{k}, \tau_1 - \tau_L) G_0(\mathbf{k}, \tau_R - \tau_2) G_0(\mathbf{k} - \mathbf{q}, \tau_2 - \tau_1) \\
&\quad \times \tilde{D}(\mathbf{q}, \tau_2 - \tau_1) d\tau_1 d\tau_2 \frac{d^3 q}{(2\pi)^3}, \\
P(X \to Y) &= p_{\text{INS}} \frac{d\tau_1}{\tau_R - \tau_L} \omega_{ph} e^{-\omega_{ph}(\tau_2 - \tau_1)} d\tau_2 \frac{e^{-\frac{q^2}{2m}(\tau_2 - \tau_1)} d^3 q}{(2\pi m/(\tau_2 - \tau_1))^{3/2}}, \\
P(Y \to X) &= p_{\text{REM}} \frac{1}{N_{\text{arcs}} + 1}.
\end{aligned}
\tag{28}
$$

Here, $p_{\text{INS}}$ and $p_{\text{REM}}$ are the probabilities to select the INSERT and REMOVE probabilities, respectively. Note in particular that all differentials cancel in the acceptance factor $r$. The reader notices that further cancellations occur in the acceptance factor such as the electron propagators between $\tau_1 - \tau_L$ and $\tau_R - \tau_2$, as well as any $\mu$-dependence. Those cancellations are only exact if function calls are used; for tabulated objects in combination with interpolation techniques there are tiny deviations from these cancellations.

*REMOVE* – This is the reverse update of INSERT. We uniformly select a phonon arc and check if its vertices are consecutive elements in the time ordered configuration (see $P(Y \to X)$ above). If this is not the case, the update is rejected. The acceptance factor is the inverse of the one determined above for the INSERT update.

*SWAP* – The INSERT and the REMOVE update allow to change the expansion order but are insufficient to generate all possible topologies because they do not allow phonon arcs to cross. The SWAP update allows one to change the topology within a given expansion order $n \geq 2$. With the notation of Fig. 7 we randomly select a vertex excluding the last one. If it has a time $\tau_1$ and the next one a time $\tau_2$, we attempt to swap the end points of their respective phonon propagators. In order to conserve momentum at every vertex, the momentum of the electron propagator between $\tau_1$ and $\tau_2$ changes – in line with our design criterion of finding updates that are local. The acceptance factor is given

by $r = W(Y)/W(X)$ with $W(X) = -G_0(\mathbf{k}, \tau_2 - \tau_1)\tilde{D}(\mathbf{q}_1, |\tau_1 - \tau_A|)\tilde{D}(\mathbf{q}_2, |\tau_2 - \tau_B|)$ and $W(Y) = -G_0(\mathbf{k}', \tau_2 - \tau_1)\tilde{D}(\mathbf{q}_2, |\tau_1 - \tau_B|)\tilde{D}(\mathbf{q}_1, |\tau_2 - \tau_A|)$.

*EXTEND* – Although this update is not needed for ergodicity, it is a useful one to improve the sampling. It changes the duration of the rightmost electron propagator in a similar fashion as the CHANGE-TAU update.

## 3.3 Implementation

The number of diagrams grows as $(2n-1)!! = (2n-1)(2n-3)\ldots$. Only the lowest expansion orders can be evaluated explicitly, but the Monte Carlo algorithm manages to sample over the most important contributions in any order. The parameter $\mu$ requires some finetuning: Its magnitude needs to be sufficiently large such that the full Green function decays exponentially. The closer $\mu$ is chosen to the unknown polaron energy $E_0$ (*i.e.*, $|\mu| \gtrsim |E_0|$) the less rapid this decay will be and the more accurate the fit (cf. Eq. 21) can be performed. For sufficiently strong $\alpha$, and $\mu$ chosen closely to $E_0$, the expansion order can be 100 or more.

Other authors prefer the use of a cyclical implementation [13] instead of a backbone line. The aim is to treat the electrons and the phonons on equal footing. It is also the structure that naturally arises at finite temperature. At zero temperature, we see little advantages for polaron problems and have not used cyclical diagrams in our codes.

## 3.4 Data structure

Let us now discuss the data structure. There are various equivalent ways to store the diagram. E.g., one may either (i) store the intervals between the emission and absorption of a single phonon along with its momentum, or (ii) one opts to store the vertices. We choose the latter approach. The necessary information needed to specify a vertex are its time, a pointer to the vertex that it connects to via the phonon propagator, the phonon momentum and at least one momentum interacting at the vertex such that all momenta can be inferred from momentum conservation. If we choose, say, to store only the phonon momenta, all electron momenta in the diagram can be computed from the given external electron momentum and by invoking momentum conservation at every vertex, but this is obviously a costly operation scaling linearly with the number of vertices. In the present implementation we decided to redundantly store all three momenta at each vertex for reasons of simplicity and memory-locality. A configuration is then specified by a time-ordered collection of such vertex objects.

When choosing the data structure, one should be conscious of the operations required by the update scheme and their respective complexity. Obviously, the ability to INSERT and REMOVE vertices efficiently while retaining the time ordering as well as the ability to seek forward and backward along the electronic backbone line are crucial, thus ruling out plain contiguous array-like data structures. Likewise, the INSERT update needs to randomly pick an electron backbone segment, the REMOVE update randomly picks a phonon propagator, and the SWAP update randomly selects a pair of adjacent vertices. All three of these ultimately draw a vertex uniformly from the set of all vertices (or in case of SWAP from all but one).

We implemented a number of different data structures to meet these requirements to varying degrees and gauge their impact.

1. A doubly-linked list as provided in C++ by `std::list` satisfies the first criterion with $\mathcal{O}(1)$ insertion and removal but requires one to start at the beginning and iterate through the list to reach a randomly picked vertex, thus resulting in $\mathcal{O}(N)$ scaling (with $N$ the number of vertices).

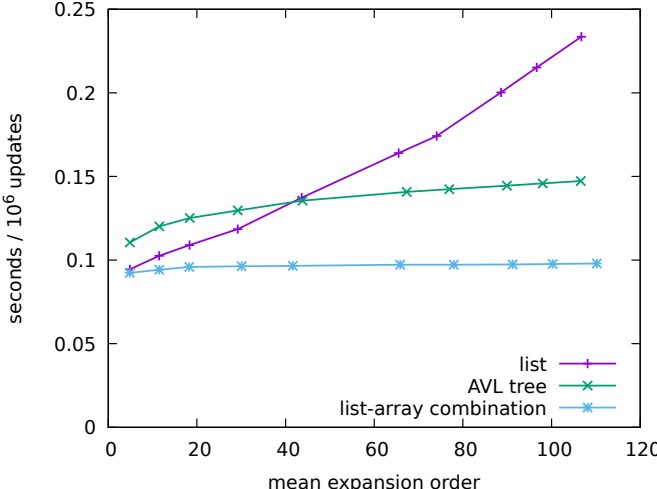

Figure 8: Benchmark of different diagram data structures. To provoke large expansion orders, $\alpha = 1$ has been used and imaginary times reached as far as $\tau_{\max} = 250$. The chemical potential has been tuned from very close to the polaron energy, $\mu = -1.02$ (high expansion order), to $\mu = -1.2$ (low expansion order).

2. A self-balancing binary search tree, e.g. an AVL or red-black tree, provides $\mathcal{O}(\log(N))$ insertion and removal and in principle also allows for true random access of an ordered sequence in $\mathcal{O}(\log(N))$ when nodes keep track of the number of nodes in their subtrees. Search trees will automatically enforce ordering which we however do not benefit from as the update scheme is designed in a way that retains time ordering anyway. While `std::map` is usually implemented in terms of binary search trees, it cannot be used off-the-shelf here as it hides its tree implementation and does not allow for the kind of additional bookkeeping required to achieve fast random access. For testing, we implemented an AVL search tree with a function to randomly access elements by index.

3. A doubly-linked list may be combined with a contiguously stored array (a `std::vector`) of iterators to the list elements that serves as a lookup table. Upon insertion, an iterator to the newly created list element is pushed to the back of the array. The list element is likewise tagged with the index of its iterator in the array. When removing a list element, its iterator in the array swaps places with the last one (updating the tag of its list element) before it is popped. This procedure retains the $\mathcal{O}(1)$ complexity of insertion and removal operations and keeps an up-to-date array containing iterators to all the list elements contiguously, albeit not in time order. Thus, we do not get proper random access but gained the ability to pick a random element in $\mathcal{O}(1)$. Care has to be taken when applying this to the SWAP update.

The performance impact of the choice of data structure depends on the average order that is reached in the course of the simulation which in turn depends on the system parameters. In our benchmark, Fig. 8, we decided to keep $\alpha = 1$ fixed and vary the chemical potential to probe a number of mean expansion orders.

In situations where the average order was below 10, the search tree (implemented as an AVL tree) performed badly compared to the list-based data structures due to the added overhead. It would only become a feasible alternative outperforming the plain list when orders beyond 40 were reached as can be seen from Fig. 8. In contrast, the list-array combination barely shows any scaling with the diagram order and was consistently faster than the plain list indicating that the overhead added due to the lookup array is very light. For models with

a sign problem where only low expansion orders can be reached, it does not matter how the data structure is implemented.

## 3.5 Error bars

The estimation of the error bars on the Green function is complicated by the fact that the normalization itself is estimated from the same simulation. We employ the jackknife resampling technique to account for that. This requires knowledge of the time series. Sampling after every single update would result in excessive memory demand and post-processing time due to many highly-correlated samples and negate the efficiency of the local update scheme. Thus, we group updates into bunches of $N_{\text{loop}}$ elementary updates. After each elementary update, we increment histograms for the Green function, the zeroth order counter used for normalization, etc. After $N_{\text{loop}}$ elementary updates, the histograms are measured, i.e. recorded in the time series, and subsequently reset. The choice of $N_{\text{loop}}$ can be guided by the estimated autocorrelation time obtained from a binning analysis.

Within the framework provided by the ALPSCore library (cf. Sec. 6), we chose to rely on the `FullBinningAccumulator` to perform the above binning analysis for us. Further, any derived quantities calculated from the observables are automatically resampled using the Jackknife method.

## 3.6 Results

For $\alpha = 1.0$, $\mu = -1.2$ and a runtime of about 1 minute on a single core laptop one can extract the polaron energy with an accuracy better than a percent. Stronger couplings are a little bit harder to simulate: We show the Green function for $\alpha = 5$, $\mu = -6$ and zero momentum in Fig. 9a. By fitting the exponential tail according to Eq. 21, one can extract the polaron energy ($E_0 = -5.55$) and residue ($Z = 0.032$). Here, the fit took data from $\tau_{\text{fit}} = 5$ onwards into account. However, for $\mu = -6$ the Green function decays rather quickly. This limits the maximum time that can be accessed with reasonable accuracy. Additionally, when taking the error bars into account in the fit, the data close to $\tau_{\text{fit}}$ impact the fit more strongly due to their higher accuracy. This leads to a heightened sensitivity towards systematic errors from the non-asymptotic fast initial decay with respect to the choice of $\tau_{\text{fit}}$.

In order to get a more reliable estimate of the polaron energy, we tuned the chemical potential to achieve longer imaginary times along with a less severe growth of the error bars. Choosing $\mu = -5.6$ (Fig. 9b), $\tau = 40$ was accessible, allowing us to probe the asymptotic regime over time scales many times that of the initial fast decay. The inset in Fig. 9b displays results for the polaron energy estimated from fits with different $\tau_{\text{fit}}$. This has been done for the data from each of the 28 independent MPI processes to yield an error on that estimate. It can be seen that after an initial influence from the non-asymptotic onset, the results beyond $\tau_{\text{fit}} = 5$ stay consistent within their error bars. Our final result reads $E_0 = -5.5498 \pm 0.0021$ and $Z = 0.03215 \pm 0.00084$. The polaron energy was thus found with a relative accuracy of $0.04\%$ at modest computational effort.

The polaron energy is remarkably close to the value predicted by Feynman's variational ansatz despite the rather strong coupling $\alpha = 5$. Feynman's trial action is parametrized by parameters $v$ and $w$, the latter of which was assumed to have only a mild influence on the end result [26]. Feynman then optimized for $v$ at fixed $w$, treating parts of the integrals approximately. From the expressions he gave in the strong-coupling regime, we find $E_0 = -5.33$ (for $w = 1$) and $E_0 = -5.39$ (for $w = 3$). With today's readily available numerical integration and optimization tools, we also optimized for $v$ and $w$ simultaneously without taking any approximations to the integrals. This results in an improved variational energy of $E_0 = -5.44$

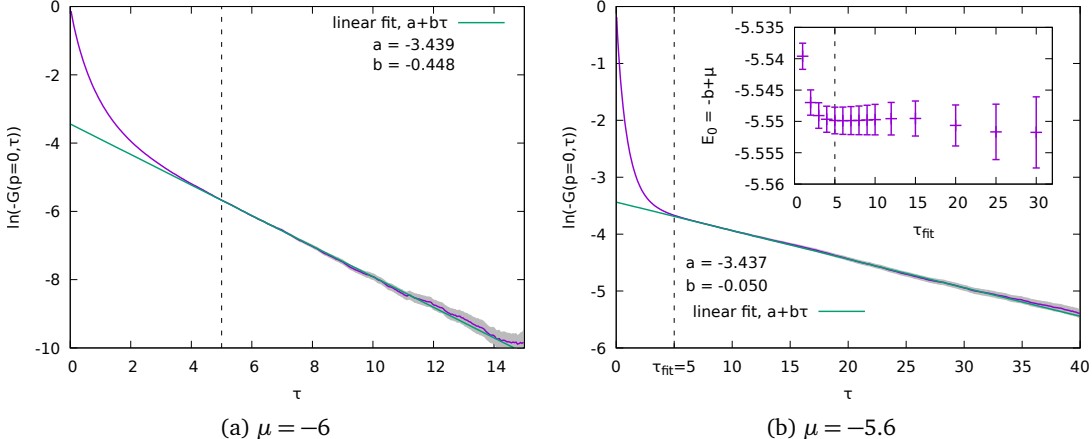

(a) $\mu = -6$  (b) $\mu = -5.6$

Figure 9: Logarithm of the Green function $\ln(-G(p = 0, \tau))$ as a function of imaginary time $\tau$ for $\alpha = 5$ and different values of $\mu$. Error bars are indicated by the shaded area. The exponential decay of the Green function is fitted starting from $\tau_{\text{fit}} = 5$ in order to obtain the polaron ground state energy and its residue, cf. Eq. 21. The inset in panel (b) shows the polaron energy (computed from the slope of the fit) for different fit windows $[\tau_{\text{fit}}, 40]$. Error bars on these data have been obtained by considering 28 independent simulations. One can see that $\tau_{\text{fit}} = 5$ is sufficiently great to probe the asymptotic regime. We switched off the CHANGE-P update and used a uniform grid in the $\tau$-direction. The code ran for 36 (a) and 467 (b) CPU-hours with the number of updates totalling $10^{12}$ and $10^{13}$ respectively.

($v = 4.03$, $w = 2.14$). Thus, our Monte Carlo estimate is 2% lower, in accordance with the variational principle.

The dispersion for $\alpha = 1$ is shown in Fig. 10. In this calculation we recalculated one of the first hallmarks of DiagMC [12]. It shows that the perturbative result incorrectly predicts an endpoint to the dispersion, whereas the DiagMC results show that the binding energy can be seen up to zero energy. In passing we note that other formula for the computation of the effective mass and the group velocity exist, see Ref. [13, 24]. The histogram over the expansion orders for $\alpha = 5$ is shown in Fig. 11. For large enough expansion orders $n$, $H[n]$ decays exponentially. The acceptance factors are about 5% for INSERT and REMOVE and 29% for SWAP. Those numbers are acceptible. If deemed too low, or if the frequency of visiting the normalization diagram is too low, reweighting and flat histogram techniques should be used.

## 4 Fröhlich polaron: self-energy

It is often advantageous to compute the self-energy instead of the full Green function and resort to the Dyson equation (Eq. 14) to obtain the latter. However, a Fourier transform from imaginary times to (Matsubara) frequencies is needed to cast the Dyson equation in algebraic form; otherwise, it is a convolution. Below we first discuss how to perform such Fourier transforms by considering the first-order diagram, and then proceed with the diagrammatic Monte Carlo computation of the full self-energy. In this text, the self-energy is always understood as the one-particle irreducible self-energy [5].

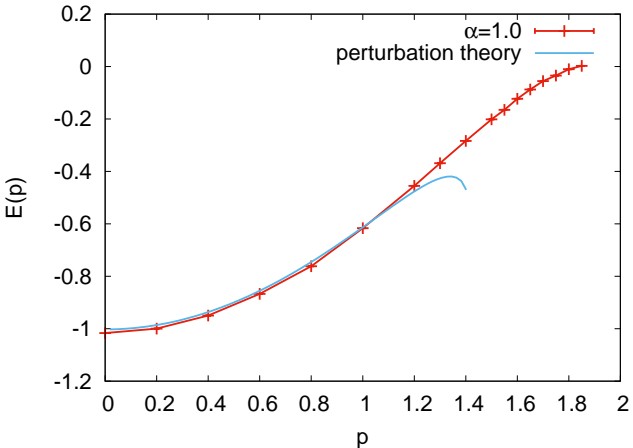

Figure 10: Dispersion $E(p)$ of the Fröhlich polaron for $\alpha = 1$ compared to the perturbative result $p^2/(2m) - \alpha\frac{\sqrt{2}}{p}\arcsin(p/\sqrt{2}) - 0.0026$ [25]. Error bars are shown but may be too small to be visible. Although the full solution shows the existence of an end point where $E(k) = 0$, the perturbative result shows unphysical behavior for large momenta. The effective mass, obtained by fitting $p^2/(2m^*)$ to the polaron energy curve for low values of the momentum $p$, is $m^*/m = 1.25$, also in reasonable agreement with the perturbative result. This calculation was historically one of the first successes of DiagMC [12].

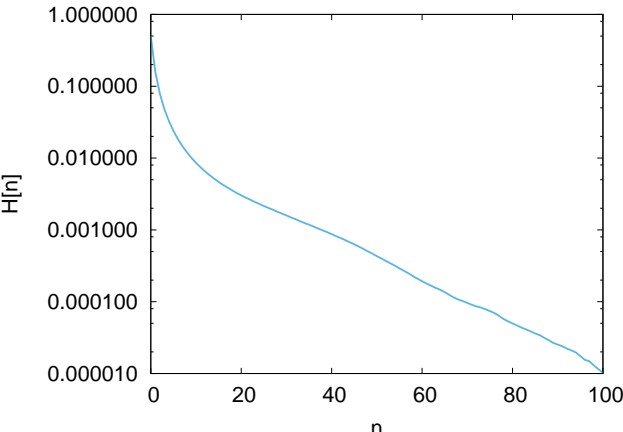

Figure 11: Histogram over the expansion orders for the same system as in Fig. 9.

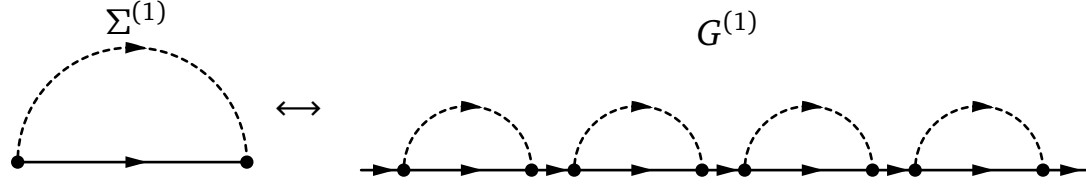

Figure 12: The first-order self-energy $\Sigma^{(1)}$ and one of the terms in the corresponding Green function $G^{(1)}$.

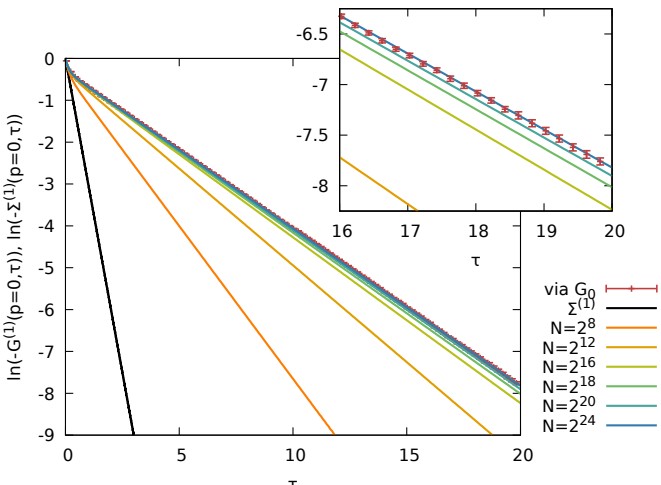

Figure 13: Computation of the Green function $G^{(1)}(p = 0, \tau)$ via the first-order self-energy. The result is shown for different number of Matsubara frequencies (powers of 2 are indicated) and compared to the result where $G^{(1)}(p = 0, \tau)$ is directly sampled (cf. Sec. 3 with minor modifications: the INSERT update is only allowed on the final electron propagator (which has zero phonon coverage) and REMOVE can only remove the last phonon arc. The SWAP update is disabled). The error bars on the latter correspond (or are smaller than) the line width. The inset shows a zoomed-in version of the last-time region. Error bars on a subset of the data sampled from the bare expansion are shown.

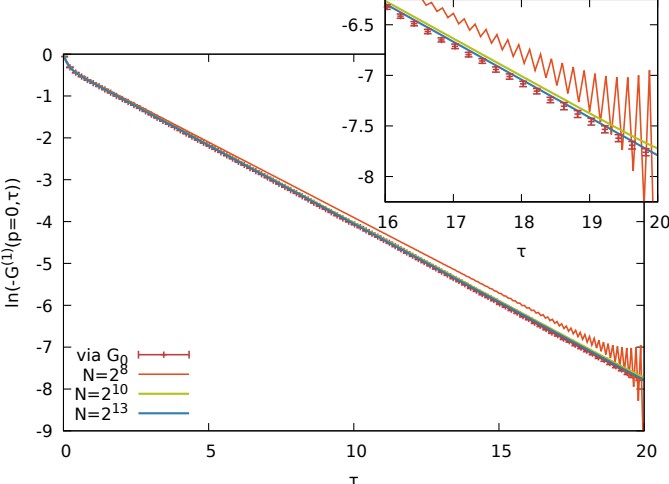

Figure 14: Same as in Fig. 13 but instead of the self-energy we compute the convolution $(\Sigma \star G_0)(\tau)$ for $p = 0$, which behaves as $\sim \sqrt{\tau}$ for $\tau \to 0$. Compared to Fig. 13, treating the divergence of the self-energy more carefully allowed us to substantially reduce the required number of Matsubara frequencies.

### 4.1 Fourier transforms explained for the first-order self-energy

The first-order self-energy is shown in Fig. 12 together with the Green function related to this diagram via the Dyson equation. The first-order self-energy can be computed analytically for zero external momentum,

$$\Sigma^{(1)}(p=0,\tau) = -\frac{\tilde{\alpha}^2\sqrt{2m}}{4\pi^{3/2}\sqrt{\tau}}e^{-(\omega_{\mathrm{ph}}-\mu)\tau}. \tag{29}$$

When applying Eq. 17 and using a root solver, we find that the polaron energy is given by $E_0^{(1)} = -2.6258286$ for $\alpha = 5$ in our units. The first-order self-energy for non-zero external momenta can be computed as,

$$
\begin{aligned}
\Sigma^{(1)}(p \neq 0,\tau) &= -\frac{\tilde{\alpha}^2 e^{-(\omega_{\mathrm{ph}}-\mu)\tau}}{p}\left(\frac{m}{2\pi\tau}\right)^{3/2}\int_0^\infty dr\,\sin(pr)e^{-\frac{mr^2}{2\tau}} \\
&= -\frac{\tilde{\alpha}^2 e^{-(\omega_{\mathrm{ph}}-\mu)\tau}}{p}\left(\frac{m}{2\pi\tau}\right)^{3/2}\left(\frac{2\tau}{m}\right)^{1/2}F\left(p\sqrt{\frac{\tau}{2m}}\right),
\end{aligned}
\tag{30}
$$

where $F$ is the Dawson function, $F(x) = e^{-x^2}\int_0^x e^{y^2}dy$. For small values of the argument, it behaves as $F(x) \approx x - 2x^3/3$. For large values of $x$, it behaves as $F(x) \approx 1/(2x) + 1/(4x^3)$. This formula for $\Sigma^{(1)}(p \neq 0,\tau)$ is the Fourier transform of the self-energy in real space $\Sigma^{(1)}(\boldsymbol{r},\tau) = G_0(\boldsymbol{r},\tau)\tilde{D}(\boldsymbol{r},\tau)$, with

$$G_0(\boldsymbol{r},\tau) = -\theta(\tau)\left(\frac{m}{2\pi\tau}\right)^{3/2}e^{-\frac{mr^2}{2\tau}+\mu\tau}, \tag{31}$$

and

$$\tilde{D}(\boldsymbol{r},\tau) = \frac{\tilde{\alpha}^2 e^{-\omega_{\mathrm{ph}}\tau}}{4\pi r}, \tag{32}$$

which can be seen as a retarded Coulombic-like potential.

In principle, all we need to do is apply the Dyson equation (Eq. 14) and Fourier transform $G^{(1)}(\boldsymbol{p},\omega)$ back to the imaginary time domain. However, as we will see, this must be done very carefully.

*Shape of the self-energy* – The most important observation is that the self-energy diverges as $1/\sqrt{\tau}$ for $\tau \to 0$ for any $\boldsymbol{p}$. This is in fact quite a common situation in continuous space, originating from the momentum integral over the bare electron Green function. The first-order self-energy very often has limits that need to be analyzed analytically. Is this divergence a problem? On the one hand, any integral $\int_0^\epsilon \Sigma^{(1)}(\boldsymbol{p},\tau)d\tau$ is convergent; in particular, there is no problem with the existence of a proper Fourier transform. On the other hand, a Taylor expansion of the self-energy around the middle of the $\tau$-bin shows that the binning process has uncontrollable systematic error bars for sufficiently small values of $\tau \to 0$. The solution is not difficult: One can choose to refrain from sampling and compute the self-energy analytically for fixed, discrete $\tau_j$ values, thereby circumventing the binning issue. If this is not an option and sampling remains essential, then one should make a measurement of $\bar{\Sigma}^{(1)}(k,\tau) = \tilde{\Sigma}^{(1)}(k,\tau)\sqrt{\tau}$, which is a featureless function of $\tau$. Whenever the discretized $\Sigma^{(1)}(k,\tau_j)$ is needed, it is obtained from $\bar{\Sigma}^{(1)}(k,\tau_j)/\sqrt{\tau_j}$. Unfortunately, this is not the only problem associated with the $1/\sqrt{\tau}$ divergence as we will see in the discussion on the Fourier transforms.

*Fourier transforms* – One of the most fundamental differences between classical mechanics and quantum mechanics is the occurrence of non-commuting operators in the latter, which in turn leads to the time ordering inherent to quantum field theory. This is already apparent from the Heaviside $\theta(\cdot)$ function in Eq. 15. It has the following frequency representation

$$\theta(t) = \int_{-\infty}^\infty d\omega \frac{e^{i\omega t}}{\omega - i0^+}d\omega. \tag{33}$$

It is this behavior which explains the structure in Eq. 16. Note that the coefficient of the $1/(i\omega_n)$ term in Eq. 16 is exactly 1 (which is identical to the jump in the Green function for $\tau = 0^+$ and $\tau = 0^-$) thanks to the (anti-)commutation relations of bosonic (fermionic) annihilation and creation operators [5]. Therefore, the same asymptotic behavior $1/(i\omega_n)$ holds not only for $G_0$ but for any Green function $G$. In the inverse Fourier transform $G(k, \omega_n) \to G(k, \tau)$, brute force summing (or integrating) over all frequencies in the hope of restoring the Heaviside $\theta(\cdot)$ function is hopeless. One therefore needs to treat the large-frequency tails carefully. At finite temperature, Fourier transforms assume that the function can be periodically continued, which is likewise violated. The easiest solution is to (i) only use the analytic formulations Eq. 15 and Eq. 16 for the bare Green function, and (ii) never perform a Fourier transform on any full Green function $G$ but only on differences $\delta G = G - G_0$. In doing so, the leading asymptotic frequencies are compensated as well as the discontinuity in imaginary time taken care of. It is also possible to treat the $1/(i\omega_n)^2$ in the same fashion: Its coefficient in frequency space corresponds to the (sum of the) slope(s) of the Green function at $\tau = 0^+$ (and $\tau = 0^-$) in imaginary time.

In the literature one can also find formulas for the $1/(i\omega_n)^3$ term, but we have never seen a case where this is necessary for the success of the calculation.

When we rely explicitly on the jump being 1 in $G$ between $\tau = 0^+$ and $\tau = 0^-$, we need to make sure in the Monte Carlo sampling that we choose $\tau_{\max}$ large enough such that $G(k, \tau_{\max}) \approx 0$ since by construction we have $G(k, 0) = -1$. As we have seen before, this means that $\tau_{\max}$ must be quite large when $\mu$ is chosen close to the polaron energy $E_0$. The region where $\Sigma^{(1)}(p = 0, \tau)$ is sizable may well appear smaller than this. Let us now use the imaginary-time and Matsubara formalisms for the Fourier transforms; specifically,

$$\Sigma(\mathbf{p}, \omega_n) = \int_0^\beta \Sigma(\mathbf{p}, \tau) e^{i\omega_n \tau} d\tau, \tag{34}$$

$$\delta G(\mathbf{p}, \tau) = \frac{1}{\beta} \sum_{n=-\infty}^{n=\infty} \delta G(\mathbf{p}, \omega_n) e^{-i\omega_n \tau}, \tag{35}$$

with the Matsubara frequencies $\omega_n = \frac{2n\pi}{\beta}$ for bosons and $\omega_n = \frac{(2n+1)\pi}{\beta}$ for fermions. Here, we have introduced a fictitious inverse temperature $\beta = \tau_{\max}$ to impose the discretization in frequency space. A single electron obviously has no statistics, so we can use either bosonic or fermionic frequencies, or just $\omega_n = \frac{n\pi}{\beta}$ – it should not matter as long as we use a transformation that turns the Dyson equation into an algebraic equation. From the Dyson equation $\delta G(\mathbf{p}, \omega_n)$ can be written as

$$\delta G(\mathbf{p}, \omega_n) = \frac{G_0(\mathbf{p}, \omega_n) \Sigma(\mathbf{p}, \omega_n) G_0(\mathbf{p}, \omega_n)}{1 - G_0(\mathbf{p}, \omega_n) \Sigma(\mathbf{p}, \omega_n)}. \tag{36}$$

Observing the decay of $G(\mathbf{p}, \tau)$ over many decades (as a function of $\tau$) requires in turn a huge number of Matsubara frequencies. The naive implementation of the Fourier transform scales as $\mathcal{O}(N^2)$ where $N$ is the number of points in the time/frequency domain and becomes too costly. Although alternative approaches exist [27], let us explain here how fast Fourier transforms (FFT) can be used, with a scaling as $\mathcal{O}(N \log N)$. For simplicity and efficiency, we rely on the open source package FFTW [28]. Although our input data is real (impying that $G(\mathbf{p}, \omega_n) = G^*(\mathbf{p}, -\omega_n)$) and we could use the function calls `r2c` and `c2r` (cf. the FFTW documentation; we could save a factor 2 in storage) we consider this advantage negligible and use instead the easier function call `dft` for complex input and output. At this point the reader should keep in mind that FFT is only a tool for solving the equations Eq. 34 and Eq. 35.

It performs

$$
\begin{aligned}
Y_k &= \sum_{j=0}^{N-1} X_j e^{-i2\pi jk/N}, \\
X_j &= \sum_{k=0}^{N-1} Y_k e^{i2\pi jk/N},
\end{aligned}
\tag{37}
$$

between input data $X$ and output data $Y$, and this is not identical to Eqs. 34 and 35. Recall what FFT really computes (cf. the FFTW documentation): first, the phase used in the forward (backward) Fourier transform corresponds to the backward (forward) sign convention in Eqs. 34 and 35; second, the forward transform immediately followed by the backward transform multiplies the input by $N$; and third, the positive frequencies are stored in the first half of the output and the negative frequencies are stored in backwards order in the second half of the output.

FFT assumes an equidistant grid where the input data are located exactly on the grid points. If we need more Matsubara frequencies than we have grid points in imaginary time, or if we use a non-uniform grid, we need interpolation methods. In practice, quadratic or spline interpolation is used. After binning the data for the self-energy, we have discretized values $\tau_j = (j + 1/2)\Delta\tau$ (in order to avoid $\tau = 0$ the same has to be done with Eq. 29); *i.e.*, we do not have the self-energy evaluated precisely on the grid points as FFT requires. If we choose bosonic Matsubara frequencies $\omega_n = 2\pi n/\beta$, then this problem does not pop up for the frequencies. Dropping the diagonal momentum index $k$ to make the notation lighter, the discretized form of Eq. 34 reads

$$
\Sigma[n] = \sum_{j=0}^{N-1} \Delta\tau \Sigma[j] e^{i\frac{2\pi n}{\beta}(j+1/2)\Delta\tau},
\tag{38}
$$

with $\Delta\tau = \beta/N$. This is almost identical to what FFT can compute for us: Apart from multiplying the input $\Sigma[j]$ with $\Delta\tau$ (the integration measure), the output self-energy of the FFT must be multiplied by $e^{in\pi/N}$ for each positive frequency $n = 0, \ldots, N/2$, and by $-e^{in\pi/N}$ for each negative frequency $n = N/2 + 1, \ldots, N - 1$. This operation needs to be undone when $\delta G(\mathbf{p}, \omega_n) \to \delta G(\mathbf{p}, \tau)$ (and we should not forget the factor $1/\beta$). In case of fermionic Matsubara frequencies, a similar phase multiplication on the data in the time domain can be derived (we leave the exact phase as an exercise).

The Green function corresponding to the first-order self-energy at zero momentum is shown in Fig. 13. We see that the required number of Matsubara frequencies is prohibitively large before agreement with the bare result is found; *i.e.,* the systematic error of the truncation in Matsubara frequencies dominates over the statistical error of the unbiased Monte Carlo sampling of the bare Green function. The reason is that the nasty $\sim 1/\sqrt{\tau}$ divergence of the first-order self-energy leads by dimensional arguments to a $1/\sqrt{\omega_n}$ behavior, which decays even slower than a Green function for large frequencies. One could treat this divergence analytically (e.g., by Taylor-expanding the self-energy), or pursue the following approach: First, notice that the $1/\sqrt{\tau}$ divergence in Eq. 30 is to leading order independent of the momentum $\mathbf{p}$ (and hence identical to the $p = 0$ result). Second, notice from the Dyson equation Eq. 36 that it suffices to compute $\Sigma(\mathbf{p}, \omega_n) G_0(\mathbf{p}, \omega_n)$ corresponding to the convolution $(\Sigma(\mathbf{p}) \star G_0(\mathbf{p}))(\tau) = \int_0^\tau \Sigma(\mathbf{p}, \tau') G_0(\mathbf{p}, \tau - \tau') d\tau'$ in imaginary time and which hence behaves as $\sim \sqrt{\tau}$ for $\tau \to 0$. This cures the divergence but still has a divergent slope: When binning data over the sampled continuous variable $\tau$ we should still measure $(\Sigma \star G_0)(\tau)\sqrt{\tau}$, as mentioned before. Before performing the Fourier transform, we subtract

$$
\delta\Sigma(\mathbf{p}, \tau) = \Sigma(\mathbf{p}, \tau) - \Sigma(p = 0, \tau),
\tag{39}
$$

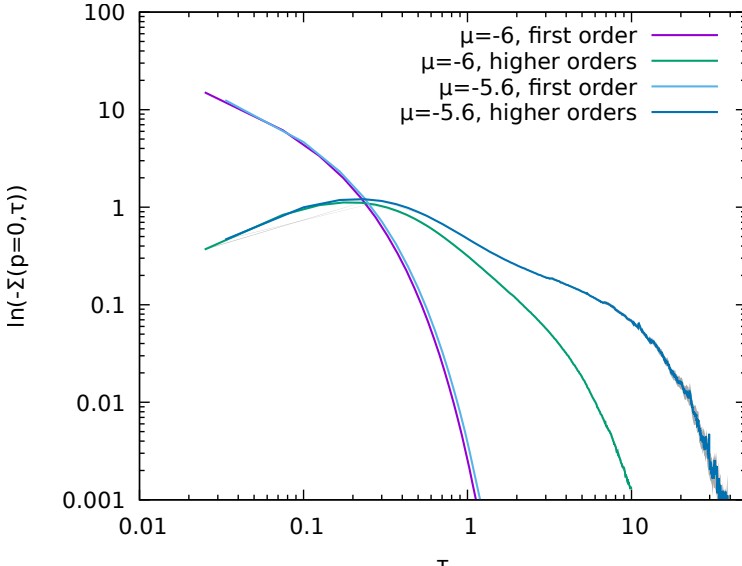

Figure 15: First order and higher orders of the self-energy $\Sigma(p = 0, \tau)$ for $\alpha = 5$ and $\mu = -6, -5.6$, respectively. Error bars are indicated by the region shaded in gray. Both simulations ran for 1330 CPU-hours each.

and compute analytically the convolution

$$\zeta(\mathbf{p}, \tau) := (\Sigma(p = 0) \star G_0(\mathbf{p}))(\tau) = -\frac{\tilde{\alpha}^2 \sqrt{m}}{(2\pi)^{3/2}} e^{-(\frac{p^2}{2m} - \mu)\tau} \int_0^\tau \frac{e^{-(\omega_{\mathrm{ph}} - \frac{p^2}{2m})\tau'}}{\sqrt{\tau'}} d\tau'. \quad (40)$$

For $p^2 < 2m\omega_{\mathrm{ph}}$ the integral is $I = \sqrt{\pi/\beta}\,\mathrm{erf}(\sqrt{\beta\tau})$, with $\beta = (\omega - \frac{p^2}{2m})$. For $p^2 > 2m\omega_{\mathrm{ph}}$ the integral is $I = 2F\left(\sqrt{\tilde{\beta}\tau}\right)e^{\tilde{\beta}\tau}\sqrt{\tilde{\beta}}$, with $\tilde{\beta} = (\frac{p^2}{2m} - \omega)$. With these manipulations the Dyson equation reads

$$\delta G(\mathbf{p}, \omega_n) = \frac{G_0(\mathbf{p}, \omega_n)(\delta\Sigma(\mathbf{p}, \omega_n)G_0(\mathbf{p}, \omega_n) + \zeta(\mathbf{p}, \omega_n)}{1 - (G_0(\mathbf{p}, \omega_n)\delta\Sigma(\mathbf{p}, \omega_n) + \zeta(\mathbf{p}, \omega_n)}. \quad (41)$$

In this approach only (a few) thousand Matsubara frequencies are needed, mostly to accommodate the decay of the Green function over many decades, see Fig. 14.

## 4.2 Computation of the full self-energy

Compared to the code for the computation of the full Green function in the bare expansion, a few minor modifications are needed for the evaluation of the self-energy $\Sigma(\mathbf{k}, \tau)$. First, the initial and final bare electron propagator have to be removed; the first vertex must have time 0 and the last one time $\tau$. We can still use a bare propagator for normalization purposes, but it obviously does not contribute to the self-energy measurement and is hence considered a fake diagram. The transition between the fake sector and the first-order diagram is best done with a separate update pair FROM-FAKE and TO-FAKE. INSERT and REMOVE allow one then to switch between expansion orders $n$ and $n+1$ for $n \geq 1$, but do not need further modifications. The EXTEND update also needs to be slightly modified because the duration of the final phonon propagator changes. Finally, in the SWAP update we need to check for one-particle reducibility: in case an electron propagator is not covered by any phonon propagator, the diagram is one-particle reducible, *i.e.*, it would fall into two pieces when cutting this propagator line. A simple

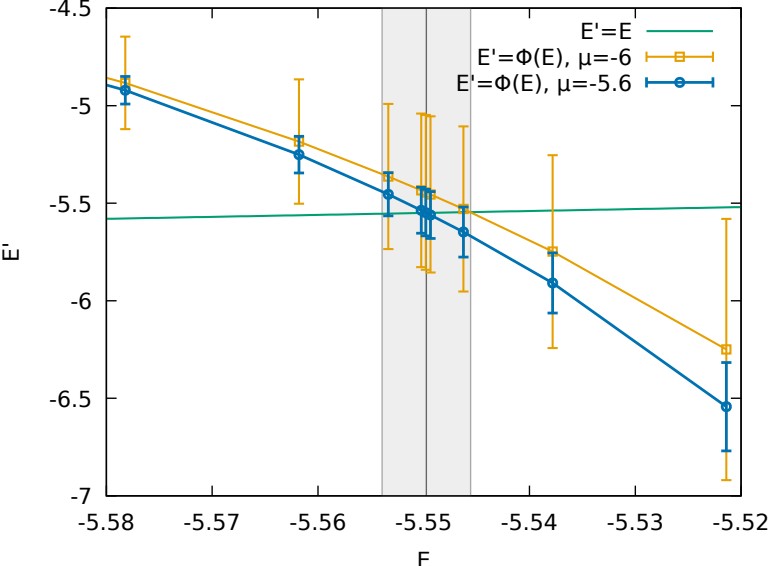

Figure 16: Energy estimation using the self-energy in imaginary time with $\phi(E)$, the right hand side of Eq. 17. Same parameters are used as before ($\alpha = 5; \mu = -6$ and $\mu = -5.6$, respectively). The intersection point with the $E' = E$ line determines the polaron ground state energy. It is marked for the case of $\mu = -5.6$ and its error bars are given by the gray area.

check for one-particle reducibility is to compare the momentum of an electron propagator with the external one. Because in the SWAP update only one electron momentum changes, this is a very cheap test to perform. The required changes to the code are left as an exercise for the reader. Note that, although the above is sufficient, the actual implementation that accompanies this document uses elements (such as DRESS-VERTEX) of Sec. 5.3 for efficiency reasons.

The self-energy is shown in Fig. 15. One sees that the first-order self-energy diverges as $1/\sqrt{\tau}$ for $\tau \to 0$ but decays rapidly for large $\tau$. The higher order terms have a vanishing contribution at $\tau = 0$ but develop an exponential tail for large $\tau$ which is important for the energy of the polaron. This figure shows that computing the convolution of the self-energy with the bare Green function is optional for the higher order terms.

In Fig. 16 we show the estimation of the polaron energy from Eq. 17. The intersection point of the right-hand side integral $\phi(E) = \int_0^\infty \Sigma(0, \tau) e^{(E-\mu)\tau} d\tau$ with the identity line $E = E$ determines the polaron ground state energy. We have sampled $\phi(E)$ in the Monte Carlo simulation directly on a non-uniform grid of $E$-values centered around the polaron energy estimate from the analysis of the Green function in the bare expansion (cf. Sec. 3.6). One could also calculate $\phi(E)$ from the discretized self-energy in post-processing and find the intersection point using a bisection scheme, albeit without the benefit of reliable error bars.

Note that the data are strongly correlated amongst different values of $E$ resulting in a relatively smooth appearance of $\phi(E)$ whereas the error analysis reveals that the errors are indeed substantial. The integral over the first-order self-energy may be calculated analytically, yielding $\phi^{(1)}(E) = -\alpha\sqrt{m/(\omega_{\text{ph}} - E)}$, thereby treating the $1/\sqrt{\tau}$ divergence exactly, while the higher orders are integrated numerically from QMC data. However, we did not find any significant discrepancy compared to taking the full self-energy.

$\phi(E)$ should be independent of the choice of $\mu$ and, indeed, the curves calculated from simulations carried out at $\mu = -6$ and $\mu = -5.6$ coincide within their errors. However, like in the bare expansion, choosing $\mu$ close to (but below) the polaron energy significantly reduces the error as longer times can be accessed before the self-energy (Green function) decays. The

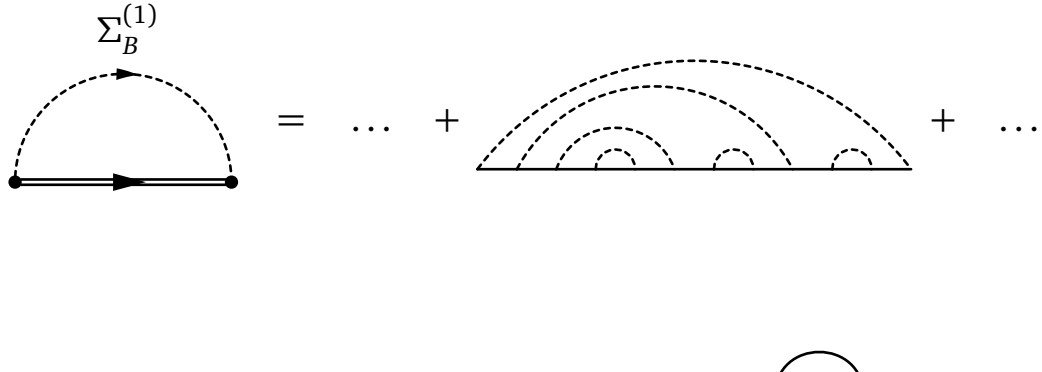

Figure 17: Upper panel: The first-order self-energy $\Sigma^{(1)}$ in the self-consistent approximation (known as the non-crossing approximation) and illustration of an included diagram in this approximation for the self-energy. The double line denotes a full electron propagator. Lower panel: the corresponding Dyson equation. In bold diagrammatic Monte Carlo one seeks a self-consistent solution to the above two equations by stochastic means and (usually) some iterative procedure.

locus of the intersection point has been interpolated. Any interpolation errors are bound to be negligible compared to the statistical one. Propagating the error onto the intersection point, we find $E_0 = -5.5497 \pm 0.0042$ or a relative error of 0.08%, in agreement with the analysis from the bare expansion, Sec. 3.6. In contrast, for $\mu = -6$, the result is $E_0 = -5.546 \pm 0.016$ which is significantly less accurate.

## 5 Fröhlich polaron: bold diagrammatic Monte Carlo

The sampling space can be further reduced when skeleton techniques are used. Graphically, this corresponds to the notion of 2-particle irreducibility: The bold diagrams for the self-energy do not fall apart when cutting any two electron propagator lines. Originally demonstrated for a (linear) scattering problem [9], it was believed that non-perturbative physics can be incorporated this way and that the series convergence could be better than for the bare series. In case the bare series is absolutely convergent, the bare and the bold series must converge to the same answer. The bold series for the Fröhlich Hamiltonian has merely a demonstrative character: in case of a convergent sign-free sampling of the bare series, it makes little sense to use anything more complicated.

### 5.1 First-order self-consistent diagram: non-crossing approximation

Let us first illustrate the method by considering the self-consistent approach to first order, *i.e.*, the diagram shown in the upper panel of Fig. 17. One sees that the self-energy depends on the full Green function $G(\mathbf{k}, \tau)$, which itself is a function of the self-energy via the Dyson equation

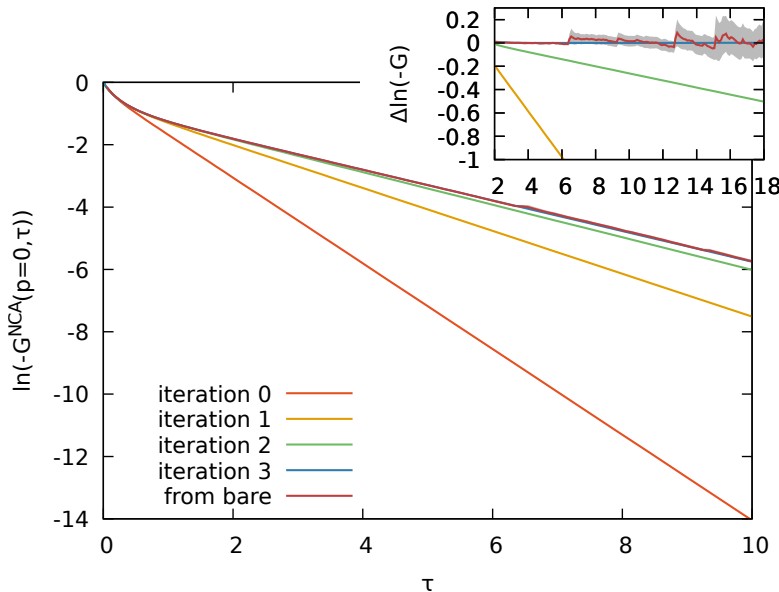

Figure 18: The Green function $G^{\mathrm{NCA}}(p = 0, \tau)$ in the non-crossing approximation for $\alpha = 5$ and $\mu = -4$. Shown are the first 4 iterations with the initial guess $G = G_0$ and compared with the result obtained by sampling the corresponding diagrams in the bare expansion. We used 256 points in imaginary time on a logarithmic grid and $2^{13}$ Matsubara frequencies. We took 200 equidistant points in momentum space with a momentum cutoff at $k_c = 100$. The results are obtained in just a few minutes on a single core. The inset shows the residues after subtracting the result after 4 iterations. The error bars on the Green function sampled from the bare expansion are indicated by the area shaded in gray.

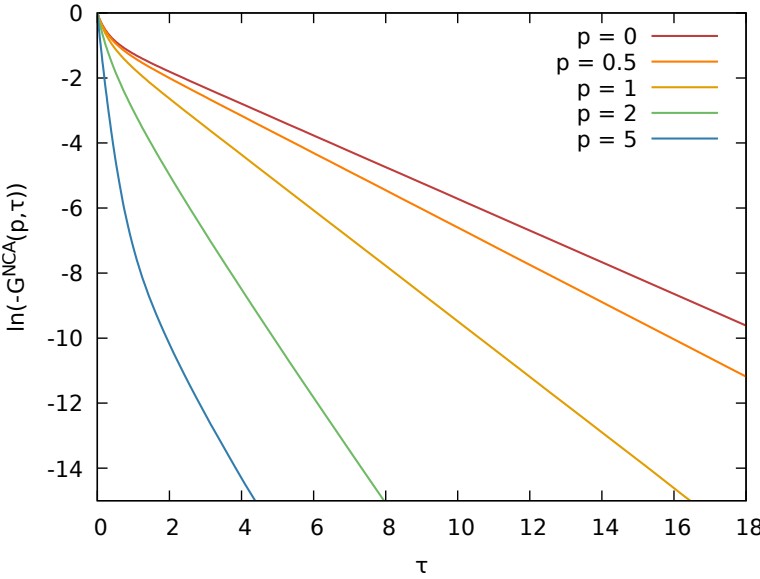

Figure 19: (Green's functions in the NCA approximation for different momenta. Same parameters as in Fig. 18.

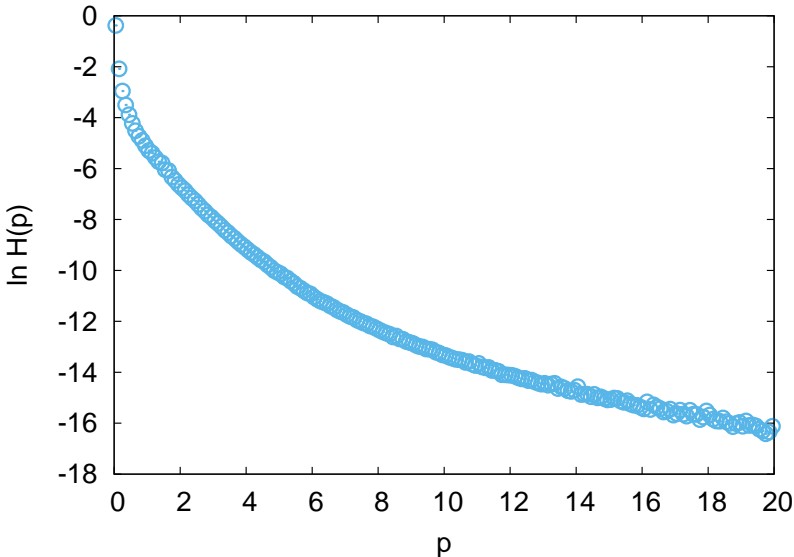

Figure 20: Histogram of the electron momenta contributing to the (full) self-energy in the bare expansion.

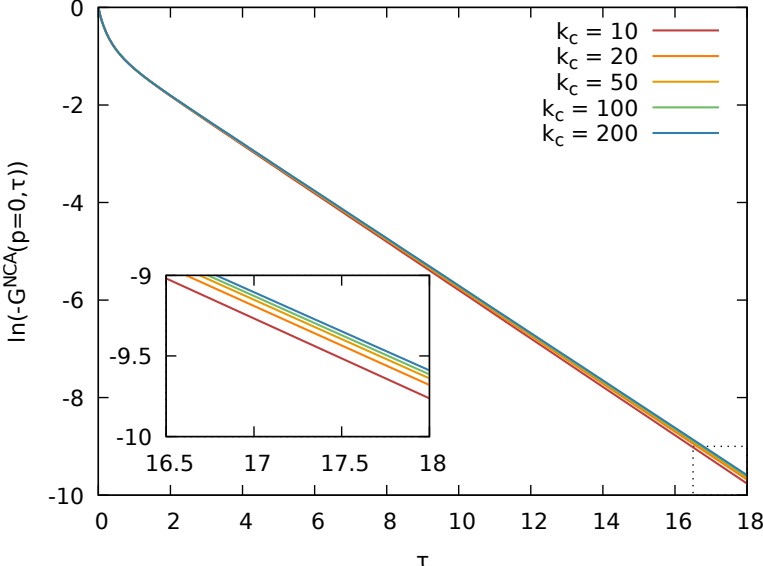

Figure 21: Dependence on the momentum cutoff for the NCA propagator. In each case, the maximum of 19 iterations has been used to ensure convergence. The inset shows a zoomed view of the same data at late times.

(see the lower panel in Fig. 17). This is a self-consistency problem,

$$\Sigma_B^{(1)}[G_B^{(1)}](\mathbf{k}, \tau) = \tilde{\alpha}^2 \int \frac{d^3q}{(2\pi)^3} \tilde{D}(\mathbf{q}, \tau) G_B^{(1)}(\mathbf{k} - \mathbf{q}, \tau), \tag{42}$$

$$G_B^{(1)}(\mathbf{k}, \omega_n) = \frac{1}{G_0^{-1}(\mathbf{k}, \omega_n) - \Sigma_B^{(1)}(\mathbf{k}, \omega_n)}. \tag{43}$$

The self-consistency problem is usually solved by iteration (note that this iteration is not a Markov process). Given an initial guess for $G_B^{(1)}$, the self-energy is computed (numerically or stochastically for the higher orders) in imaginary time and subsequently Fourier transformed to Matsubara representation, from which a new Green function is extracted via the Dyson equation and brought back to imaginary time representation by an inverse Fourier transform, and this procedure is repeated until convergence is reached. It is often needed to introduce a damping factor. As can be seen in Fig. 17 the diagrams thus summed correspond in the bare series to all possible diagrams in which the phonon lines do not cross (the so-called non-crossing approximation (NCA)).

Given the previous experience with numerical instabilities in the first-order self-energy using the bare expansion (see Sec. 4.1), we anticipate the same problem. We split hence

$$G_B^{(1)}(\mathbf{p}, \tau) = G_0(\mathbf{p}, \tau) + \delta G_B^{(1)}(\mathbf{p}, \tau),$$
$$\Sigma_B^{(1)}[G_B^{(1)}](\mathbf{p}, \tau) = \Sigma^{(1)}[G_0](\mathbf{p}, \tau) + \Sigma_B^{(1)}[\delta G_B^{(1)}](\mathbf{p}, \tau), \tag{44}$$

that is, we subtract the bare propagator from the bold propagator and evaluate the corresponding contributions to the self-energy separately. The first part is simply the first-order self-energy $\Sigma^{(1)} \equiv \Sigma_B^{(1)}[G_0]$, cf. Eqs. 29 and 30. Henceforth, we abbreviate the second part $\Sigma_B'^{(1)} \equiv \Sigma_B^{(1)}[\delta G_B^{(1)}]$ and reduce it to a one-dimensional integration,

$$\Sigma_B'^{(1)}(p = 0, \tau) = \frac{\tilde{\alpha}^2}{2\pi^2} e^{-\omega_{\mathrm{ph}}\tau} \int_0^\infty \delta G_B^{(1)}(q, \tau) dq,$$
$$\Sigma_B'^{(1)}(p \neq 0, \tau) = \frac{\tilde{\alpha}^2}{(2\pi)^2} e^{-\omega_{\mathrm{ph}}\tau} \frac{1}{p} \int_0^\infty \delta G_B^{(1)}(q, \tau) q \ln\left|\frac{p + q}{p - q}\right| dq. \tag{45}$$

The integral can be split as

$$\int_0^\infty \ldots = \int_0^{p - \Delta p} \ldots + \int_{p - \Delta p}^{p + \Delta p} \ldots + \int_{p + \Delta p}^\infty \ldots \tag{46}$$

where the first and third integral can be evaluated numerically and the middle integral vanishes in the limit $\Delta p \to 0$,

$$\lim_{\Delta p \to 0} \int_{p - \Delta p}^{p + \Delta p} \delta G_B^{(1)}(q, \tau) q \ln\left|\frac{p + q}{p - q}\right| dq$$
$$= \lim_{\Delta p \to 0} \left[ \delta G_B^{(1)}(p, \tau) p \, 2\Delta p \ln(2p) - \delta G_B^{(1)}(p, \tau) p \Delta p (2\ln(\Delta p) - 2 + i\pi) \right] = 0. \tag{47}$$

The first-order contribution $\Sigma^{(1)}[G_0]$ in Eq. 44 singles out the $1/\sqrt{\tau}$ divergence of the self-energy. As before, it should be treated carefully, for instance as the convolution $(\Sigma^{(1)}(\mathbf{p} = 0) \star G_0(\mathbf{p}))(\tau)$ which we suggested before.

The corresponding Green function is shown in Fig. 18 for zero momentum. The momentum dependence of the Green function can be seen in Fig. 19.

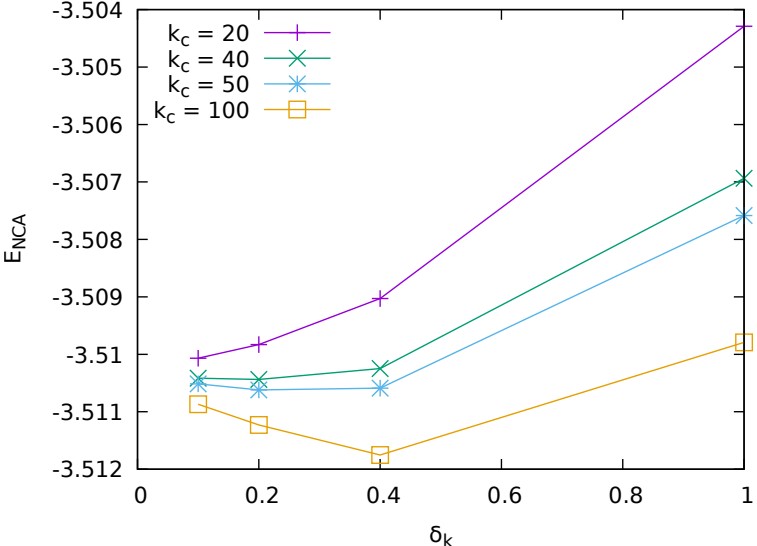

Figure 22: Dependence of the polaron energy in the NCA approximation on the momentum cutoff $k_c$ and the number of momenta $N_k$ chosen uniformly with spacing $\delta_k = \frac{k_c}{N_k}$. Other parameters are $\mu = -4$, $\tau_{\max} = 20.48$ with 512 time points chosen on a logarithmic grid, and $2^{16}$ Matsubara frequencies. The energy was determined from linear regression of the exponential tail of the Green function between $\tau = 5$ and $\tau = 18$.

## 5.2 Grid and momentum cutoff

It is seen in Fig. 17 that even the computation of $\Sigma_B^{(1)}(p = 0, \tau)$ requires knowledge of $G_B^{(1)}(\mathbf{p}, \omega_n)$ for all momenta $\mathbf{p}$. We hence need a two-dimensional grid for $G$ and $\Sigma$ in $(p, \tau)$ space, and biquadratic (or bicubic if affordable) interpolation. We also need to introduce a momentum cutoff to store the measurement of $\Sigma$. We now analyze if (and when) the momentum cutoff matters.

The bare Green function decays with momentum as a gaussian for fixed values of $\tau$. Large values of $\tau$ are usually unimportant because of the $\mu$ dependence in the propagator, providing a cutoff in time (even for $p = 0$ this term is present). If large values of $\tau$ occur, they provide a cutoff for the momenta of the order of $p \sim \sqrt{2m/\tau}$. However, for very small values of $\tau$ there is no restriction on the values that $\boldsymbol{p}$ can take.

To see what influence large momenta have in practice, we show in Fig. 20 the histogram of the logarithm of the modulus of all electron momenta contributing to the self-energy (using the bare $G_0$ expansion) when the external momentum is $p = 0$. The main contribution is at low momenta, as expected, and large momenta are suppressed as a power law with exponent 4.25(5); contributions for $p > 4$ are seen to be already very small. But given that our requirement on precision is extremely high (just recall the reported very small values of the Green function for large values of $\tau$ – surely we need our systematic error to be much smaller than the signal), it is not a priori clear that the cutoff dependence is going to be negligible.

This is likewise reflected in the first-order (non-bold) self-energy. Choosing a very small $\tau$, we see that for $p^2 \ll 2m/\tau$ the first-order self-energy is to leading order momentum independent (namely, a large constant because of the $\sim 1/\sqrt{\tau}$ divergence) but for $p^2 \gg 2m/\tau$ it behaves as $\sim p^2$. This follows from the asymptotic expansions of the Dawson function. The momentum dependence in the NCA approximation is unfortunately not identical to the one in the bare first-order self-energy and hard to grasp analytically.

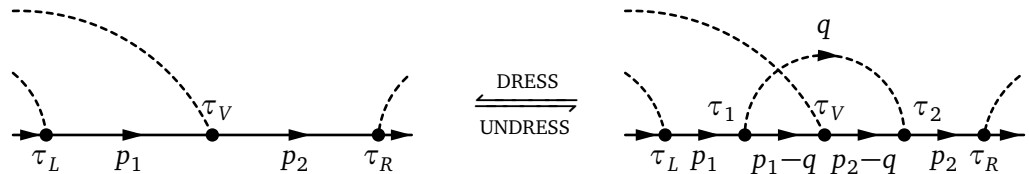

Figure 23: Illustration of the VERTEX-DRESS / VERTEX-UNDRESS pair of updates.

Higher order diagrams should be better behaved: When phonon lines cross, then phase space arguments for $\tau$ show that small values of $\tau$ are not giving important contributions to the self-energy. We believe it is indispensable to numerically check the influence of the cutoff, which we show for the Green function in the NCA approximation in Fig. 21. Note that we used bare propagators whenever momenta whose magnitude is higher than the cutoff were requested, but it makes little to no difference compared to a hard cutoff. This is an approximation but would lead to a correct evaluation of the first-order bare diagram in all circumstances. It is seen in Fig. 21 that the cutoff dependence is not pronounced.

However, quantities such as the energy converge rather slowly with the cutoff parameter (since the energy corresponds to the asymptotic decay of the Green function one can appreciate this aspect from Fig. 21) even though the approximate values are very close to the final one. Furthermore, precise energies remain sensitive to the discretization. Getting control beyond the $\sim 0.5\%$ level of accuracy on such quantities is not an easy task in a self-consistent approach, and a systematic study of all parameters is warranted. We show a characeristic example in Fig. 22, where we modify the cutoff as well as the momentum spacing. We chose the time discretization such that it is about two orders of magnitude weaker than the momentum discretization in this plot. However, choosing a smaller starting value for the fitting interval over which the energy is determined leads to fluctuations of the same order of magnitude as in Fig. 21. Since the tail should be fitted for the energy, we believe that our starting value ($\tau_{\text{fit}} = 5$) is conservative. A striking feature in Fig. 22 is the non-monotonous behavior for large cutoffs, indicating that too few momenta might well have led to erroneous extrapolations. At the moment we see no other option than a purely numerical analysis of this dependence, and argue that it is indispensable to perform such checks.

## 5.3 Code

Bold DiagMC requires only a couple of changes to the code for the self-energy:

*Boldification* – This step has been described already in Sec. 5.1.

*New updates* – The current implementation of the INSERT update automatically leads to a two-particle reducible diagram. One possibility is to keep the updating scheme as is supplemented with introducing a flag signalling two-particle reducibility, and making sure that the self-energy is measured only in the irreducible space. There exists however a way to add a phonon arc such that it always leads to an irreducible diagram. It works as follows (see Fig. 23): first a random vertex at time $\tau_V$ is chosen, excluding the one at $\tau = 0$. Then we propose to insert a new phonon propagator that dresses this (but only this) vertex. This is also a local update, and since it increases the number of phonon line crossings by one, this update always leads to a physical diagram. We leave the derivation of detailed balance for this VERTEX-DRESS/VERTEX-UNDRESS pair as an exercise, but note that choosing the last vertex requires special care.

*Irreducibility checks in SWAP* – In a bold code we need to make sure that no subpiece of a diagram can be identified with a lower order diagram already taken into account (which is the same as the requirement of two-particle irreducibility). Fortunately, there exists a simple

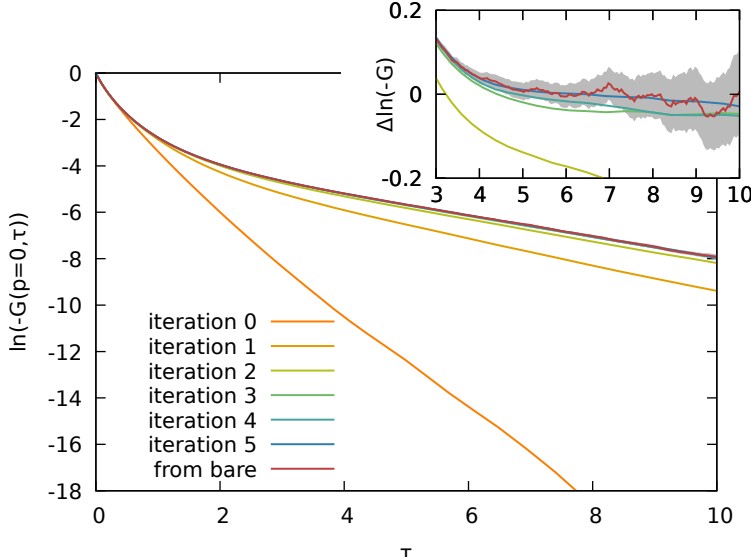

Figure 24: The Green function $G(p = 0, \tau)$ obtained in the bold approach for $\alpha = 5$ and $\mu = -6$. Shown are the first six iterations (bottom to top, obtained by sampling) with the initial guess $G = G_0$ and compared to the result obtained in the bare expansion. Error bars on the latter are indicated by the shaded region. We used 256 points in imaginary time on a logarithmic grid and $2^{13}$ Matsubara frequencies. We took 200 equidistant points in momentum space with a momentum cutoff at $k_c = 100$. The inset shows the same data with the asymptotic line fit from Fig. 9a subtracted, $\Delta \ln(-G) = \ln(-G) - (a + b\tau)$.

check: if no 2 momenta are identical then the diagram is bold irreducible. If one uses the VERTEX-DRESS/VERTEX-UNDRESS updates, then reducibility can again only happen during the SWAP update, and one only needs to check the new momentum $\mathbf{k}'$ between $\tau_1$ and $\tau_2$ against the incoming and outgoing momenta of $\tau_A$ and $\tau_B$ (for the notation see Fig. 7). If this simple check cannot be done (as in any system not dealing with polarons), then one keeps track of the momenta in the diagram in the form of a hash table. This allows for a quick check if a new momentum is suggested. Momenta whose values change are removed from the table and the new ones are added.

## 5.4 Results

For our standard example $\alpha = 5$ we see in Fig. 24 that the full Green function agrees with the one obtained from the bare expansion (cf. Fig. 9). We also see that only few iteration steps are needed before convergence is reached. In particular, the inset reveals that after six boldification iterations, it stays well within the error bars of the bare Green function over the whole range of imaginary times considered. We do not have any rigorous means of obtaining error bars on the Green function in the bold scheme, however the norm of the change $\delta G_B$ due to boldification may be considered indicative of convergence. Since we do not reset our observable for the self-energy after boldification but just keep on sampling with respect to the new bold propagator, we have an easy way to systematically improve our results and diminish the statistical Monte Carlo errors by just carrying out further bold iterations. Otherwise, one would need to employ some sort of heuristic for increasing the number of Monte Carlo sweeps within each iteration step to reflect the higher demand on statistical accuracy as self-consistency is reached. If the desired accuracy is not reached a comprehensive analysis of the sources of all systematic errors is necessary, which combined with the statistical errors from the sampling can be a cumbersome

task.

The data presented in Fig. 24 took 840 CPU-hours (30 hours on one 28-core Broadwell node @ 2.4 GHz) to gather. Each of the six iteration steps consisted of $7.5 \times 10^{12}$ elementary Monte Carlo updates. The time taken by the actual boldification step is negligible in comparison, at only a few minutes each. The bold scheme thus requires significantly more computational effort to achieve results that are comparable to the bare approach. However, one should be mindful that the bold scheme has a somewhat larger configuration space to cover as it samples the self-energy at different momenta while the momentum was kept fixed at zero in Fig. 9.

# 6   Open source codes

We provide our C++ implementations of the DiagMC method for the systems discussed in parts 2 through 5 under an open source license (GPL v3). They are available through the Git repository at https://gitlab.lrz.de/Lode.Pollet/LecturesDiagrammaticMonteCarlo .

Our codes make use of the ALPSCore library [29], based on the original ALPS project [30]. ALPSCore employs the HDF5 data format [31], as well as the Boost C++ libraries [32]. Further, we rely on the FFTW3 library [28] for the Fourier transform necessary for the Dyson equation in the self-energy formalism. Finally, the Faddeeva package implementation of the Dawson function [33] is used in the calculation of the first order of the self-energy.

Please refer to the README files in the code repository for technical details on how to build and run the codes. We also provide parameter files which allow you to reproduce the data depicted in Figs. 9, 15, 16, and 24.

# 7   Outlook

In these notes we only discussed the concepts of the (irreducible) self-energy and skeleton diagrams for the Green function propagator. In a many-body context, the (irreducible) polarization and the effective interaction can be treated in the same way and give rise to such effects as screening (a well-known example is the electron gas model [5]). Graphically, the interaction is also a two-point line-object: The interaction corresponds to the propagation of a single boson. For polaron and impurity-like problems, the medium is considered an infinite bath and can hence not be renormalized. In practice, bold DiagMC schemes rely on the $G^2W$ scheme [34].

More generally bold diagrammatic elements can also be introduced at the two particle level. The full system of non-perturbative self-consistent equations are known as the Hedin equations [35]. The central object of the 5 Hedin equations is the 3-point irreducible vertex; Green functions and effective interactions are related via their respective Dyson equations to the self-energy and the polarization, whereas the vertex can be expressed in terms of bold propagators and the irreducible vertex. There is however no closed form for the right hand side in the self-consistent equation for the 3-point vertex (in the language of functional integrals, it is possible to write down the right hand side as a functional derivative, but this remains impractical for an actual numerical computation). Thus far, the self-consistent treatment of the 3-point vertex has not been attempted in diagrammatic Monte Carlo because of the curse of "dimensions": already for the Fröhlich polaron in 3 dimensions with rotational symmetry it is a 5-dimensional object whose storage, interpolation and stochastic evaluation are non-trivial.

# 8 Extensions

In this final section we very briefly discuss a number of systems that can rather straightforwardly be studied with the techniques outlined in this manuscript. Our goal is to show the similarities between these systems from the algorithmic point of view rather than a full discussion of the physics of these models, which is beyond the scope of these lecture notes. Wherever possible, we will provide references to reviews.

## 8.1 Acoustic phonons

In contrast to the optical branch relevant for the model discussed in the main part of the text, acoustic phonons have a linear dispersion, $\omega_{\mathbf{q}} = cq$ with $c$ the speed of sound. The derivation of the large polaron in the Fröhlich model results in a coupling $V(q) \sim \sqrt{q}$ between the impurity and the phonon bath [36]. Consequently, a UV-cutoff is needed as can easily be seen by writing down the first-order self-energy expression. The polaron properties were computed using diagrammatic Monte Carlo in Ref. [37]. Just as for the optical phonons, Feynman's variational Ansatz [26] is in excellent agreement with the full results, but predicts a transition between a quasi-free and a self-trapped polaron, which may be continuous or discontinuous depending on the value of the cutoff [36]. This transition was also observed in a path-integral Monte Carlo study [38], seen as a jump in the potential energy. It cannot be ruled out that this jump is cutoff dependent, and this jump is not a proof of any (strong or weak) localization. The authors of Ref. [37] did not elaborate but noted that the structure of the diagrams which contribute significantly differs in the quasi-free and self-trapped regimes.

## 8.2 Bose polaron

The Bose polaron describes an impurity immersed in a weakly interacting Bose-Einstein condensate (BEC). The system is usually modelled with $\delta$-pseudopotentials in continuum space, after which the interactions of the BEC are linearized using the Bogoliubov prescription, resulting in a Fröhlich type of Hamiltonian, with phonon dispersion $\omega(k) = kc\sqrt{1 + (k\xi)^2/2}$, where $c$ is the speed of sound and $\xi$ the healing length of the BEC. The coupling of the impurity to the BEC phonons is given by $V(q) \sim \left(\frac{(\xi q)^2}{(\xi q)^2 + 1}\right)^{1/4}$. This derivation breaks down however when the impurity is able to sufficiently deplete the condensate: As soon as the impurity gets dressed by two (or more) phonons, one must also take the full density-density repulsion of the bosons into account for stability reasons, but this lies outside the Fröhlich BEC polaron model. One therefore expects substantial differences between experiments and the predictions of this model in the strongly interacting regime. An excellent review of the physics of the Bose polaron can be found in Ref. [39]. The Fröhlich-type BEC polaron model is, just like the previously discussed acoustic phonons, UV divergent and a renormalization scheme is essential. In the diagrammatic Monte Carlo study of Ref. [37] it was found that the momentum cutoff had to be orders of magnitude larger than any physical parameter before the (renormalized) ground state energies could be reliably extrapolated in the inverse of the cutoff. Fluctuations over many orders of magnitude make the simulations inefficient. It would hence be interesting to revisit this problem with a different renormalization scheme, and/or utilizing a partial resummation of diagrams to cure the sensitiveness to the cutoff.

## 8.3 Fermi polaron

When an impurity is immersed in a dilute, non-interacting Fermi sea, the ground state can either be a polaron or a molecule when the impurity forms a bound state with just one fermion.

Like for the Bose polaron, the interactions between impurity and bath originate from a typical cold atom setup with all their benefits: The scattering lengths can be tuned, even made infinitely strong, but the interactions remain of zero range. The Fermi polaron was one of the first hallmarks of modern diagrammatic Monte Carlo simulations [40, 41], firmly establishing the polaron-to-molecule transition. Note that the presence of fermionic propagators leads to a sign-problem, which makes the simulations much harder than for bosonic problems and limits the reachable expansion orders typically to 8-12, depending on the dimensionality, interaction strength, species mass, etc [42–45]. A particularly elegant way to deal with the UV divergence and resonant interactions simultaneously is by introducing the T-matrix [40, 41]. There exist excellent reviews on the topic of the Fermi polaron, such as Refs. [46, 47].

## 8.4 Multi-polaron systems

A finite density of electrons coupled to optical phonons within the Holstein model (*i.e.,* the electron density couples locally to the displacement operator via a coupling of the form $g \sum_i a_i^\dagger a_i (b_i^\dagger + b_i)$ where the amplitude of the coupling is $g$ and $a_i$ the electron/impurity annihilation operator on site $i$ and $b_i$ the phonon one) was considered in Ref. [48]. In this case, the presence of a Fermi surface for the electrons leads again to a sign problem (since the particle and the hole propagators have opposite sign). In Ref. [48] the bold series was sampled up to orders 4-6, high enough to observe convergence. It was found that the effective mass increases and the residue decreases with increasing electron density at fixed coupling strength for typical metals. The authors found that approximating the self-energy with a purely local one is accurate to 2%.

## 8.5 Spin-boson models

The spin-boson Hamiltonian is the prototypical model for a quantum-mechanical system embedded in a dissipative bath [49], describing the coupling of a two-level system to an infinite bath. It is defined as

$$H = \Delta \sigma_x + \sigma_z \sum_i \lambda_i (b_i^\dagger + b_i) + \sum_i \omega_i b_i^\dagger b_i, \tag{48}$$

where $\sigma_x$ and $\sigma_z$ are the Pauli spin-1/2 operators, $b_i$ and $b_i^\dagger$ are boson creation and annihilation operators, $\omega_i$ are harmonic oscillator frequencies, and $\Delta$ is the tunneling matrix element. The coupling between the spin and the bosonic bath is determined via the $\lambda_i$ by the spectral function $J(\omega) = \pi \sum_i \lambda_i^2 \delta(\omega - \omega_i) = 2\pi \alpha \omega_c^{1-s} \omega^s$ for $\omega < \omega_c$ and zero otherwise. The parameter $\alpha$ describes the coupling strength to the dissipative bath. The parameter $s$ distinguishes between a sub-ohmic bath ($s < 1$) and an ohmic bath ($s = 1$). At zero temperature and for $s \leq 1$ the system undergoes a phase transition at finite coupling strength $\alpha_c$ between a delocalized and a localized state, in which the system is no longer able to tunnel and behaves essentially classically. There was controversy about the nature of the phase transition in the sub-ohmic case. On general grounds one expects the transition to fall in the universality class of the classical Ising model with long-range interactions with mean-field critical exponents for $s < 1/2$. The first numerical group renormalization studies observed however different exponents obeying hyperscaling for $s < 1/2$ and argued the breakdown of the quantum-to-classical mapping.

The continuous time Monte Carlo simulations of Ref. [50] are free of systematic errors and could establish the exactness of the quantum-to-critical mapping by observing the expected mean-field exponents. The discrepancies had thus to be found in the truncation of the bosonic Hilbert space in the numerical renormalization group approach. The Monte Carlo sampling of

this system resembles Sec. 2 but needs to be augmented with a cluster update for the retarded spin-spin interactions, see Ref. [50], resulting from integrating out the bath modes.

For a comprehensive review of the physics of spin-boson models, see Ref. [51].

## 8.6  Anderson localization

When free fermions can hop on a lattice subject to disorder in the chemical potential, they will always localize in 1D and 2D and for strong enough disorder in 3D. For quenched disorder drawn from a Gaussian distribution, the diagrammatic technique is simplest to derive. The diagrammatic structure is in fact very similar to Sec. 8.4: The electron propagator is dressed with arcs, but those arcs have no time-dependence (in contrast to the exponential decay for the polarons, see Eq. 13). The Green function at zero temperature on a 3D lattice was computed in real time in Ref. [52]. While unable to locate the transition (which requires the computation of the conductivity and analyzing it for low frequencies and momenta), it showed the very strong local character of the self-energy (cf. Sec. 8.4).

## 8.7  Impurity models

Models such as Anderson's impurity model occur as auxiliary problems in dynamical mean-field theory, when one seeks to sum over all skeleton diagrams for the self-energy built with purely local Green functions. The important point is that this sum is not accomplished directly but through the impurity problem, for which a variety of Monte Carlo solvers have been developed in continuous time, see Ref. [1] for a review. One expands in the interations (CT-INT), performs a Hubbard-Stratonovich decoupling of the interactions (CT-AUX), or expands in the hybridization (CT-HYB). For the bosonic impurity problem, only an expansion in the kinetic term has thus far been developed (cf. CT-HYB), see Refs. [53, 54].

## 8.8  Real-time phenomena

The spectral function [13, 24] and the optical conductivity [55] have been determined from the corresponding imaginary time correlation functions for the Fröhlich polaron using analytic continuation methods. The optical conductivity of the Holstein model was studied in Ref. [56, 57], as well as its mobility [58]. To date, no polaron studies have been published directly for real time following the approach of Ref. [52] for the Anderson model.

By contrast, impurity models have also been studied to address out-of-equilibrium phenomena, see Refs. [59–68]. One is typically interested in the transport of quantum dot like systems coupled to external leads, and attempts to monitor the time evolution for a long enough period of time such that a steady state sets in.

# 9  Conclusion

The purpose of these notes is to provide a pedagogical overview of the technical aspects of diagrammatic Monte Carlo simulations, lowering the barrier for newcomers, and giving a flavor of its power to experienced researchers acquainted with other numerical techniques. With the techniques outlined here interesting physics has been discovered and established unambiguously in the past. With only minor changes open, challenging problems can still be attacked, and we gave a number of examples in the previous section. To study the complexity of strongly interacting problems a few more steps are needed, such as resummation techniques, more updates, and sign alternations. The series will in general not be convergent, which we consider to be the greatest challenge for diagrammatic Monte Carlo simulations, and the diagrammatic

structure is more complicated than the diagrams considered here, which all have a backbone line for the impurity propagator. Just as for the Fröhlich polaron, it is imperative to treat as much as possible of the physics in an analytical way. Having gone through this tutorial the reader can understand better the technical aspects of the method, appreciate the efforts described in the literature, or start coding and exploring on their own.

## Acknowledgements

This work would have been impossible without the numerous ideas and selfless contributions of collaborators and students. This work was supported by FP7/ERC Starting Grant No. 306897 (QUSIMGAS) and the DFG through Nano-Initiative Munich.

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
