# Peer review of "Lecture notes on Diagrammatic Monte Carlo for the Fröhlich polaron"

_SciPost Physics Lecture Notes, doi:SciPost Phys. Lect. Notes 2 (2018)_

## Round 2 · Referee Report · Anonymous (Referee 1) · 2018-2-1

Strengths

1- Useful if used on a hands-on course together with numerical exercises. 2- Codes available on-line

Weaknesses

1- a little bit too technical

Report

I am satisfied with the overall form of this lecture notes. They are written clearly and it would be of help for an hands-on course where the audience can profit of available codes.

An issue is related to Eq. (17). Derivation of this relation assumes that a pole is well separated from a continuum which is not always the case for a large polaron. In particular the condition $\tau\gg\omega^{-1}_{ph}$ do not guarantee the validity of Eq. (17). For Eq. (17) to be true it is necessary that $\tau\gg\Delta^{-1}$ where $\Delta$ is the energy difference between the pole and the edge of the continuum. I expect deviations from a pure exponential behaviour as far as the pole approaches the continuum ($p$ approaches the end-point in Fig. (10)). This will affect also the errobar of the energy around this point. I think that this is a general point i.e. is characteristic of part of the spectrum where (in real frequencies) an imaginary part of the self-energy is present and possibly large, therefore I think that a more careful discussion is needed.

I found sect. 3.4 too technical. Related to the discussion in sect. 3.4 the statistics of the diagram order should be improved according to my opinion. In both sections 2 and 3 a discussion on the sampling of the order of perturbation theory will be useful to the reader. In the datasets produced for the two level system (sect. 2) seems not to provide this information. Could the authors provide this information?

I also suggest the authors to supplement the paper with a list of exercises or propose exercises whenever possible. In the present of the paper on page 23,25 and 31 there are suggestions for possible exercises. I think it would be useful to regroup it on a devoted section or emphasize better in the text.

Requested changes

In the abstract the warning about what this paper is *not* seems to be too restrictive. Mentioning the introduction and the extension of the method described in the paper it would be valuable for a reader.

Describing the slicing procedure (Fig.3) it would be of help for a reader to comment the method w.r.t the usual fixed size slicing (trotterization).

Derivation of Eq. (17) is needed (see report).

Page. 11/12 describing the perturbation theory it should be of help having a numbered formula for the phonon propagator as Eq. (17) for the electron.

After Eq. (19) the "first equation" and "second equation" shoud be substituted with a reference to Eq. (18) and Eq. (19).

In Eq. (25) but also in previous part of the paper the Weights, which are proportional to odd number of electron propagators are negative, perhaps a comment on this fact here or in the previous section should be needed.

There's a typo in Eq. (39)

Color codes and linewidth of the figures are hard to read. Changes are requested for figures 9, 13, 14, 15, 18, 19, 21, 24. Shaded grey for errorbar should be make more visible.

In sections 3.6 and 4.2 result for the ground state energy obtained with free and bod method should be compared in a table to make them more quickly accessible to the reader.

Perhaps a list of suggested exercises based on the codes (which are now proposed in the text) should be added at the end of the manuscript.

  • validity: high
  • significance: ok
  • originality: good
  • clarity: high
  • formatting: excellent
  • grammar: good

Author:  Jonas Greitemann  on 2018-03-05  [id 222]

(in reply to Report 1 on 2018-02-01)

We thank the referee for their time and effort to review our manuscript. We found ourselves in agreement with most points raised and altered the manuscript accordingly. We are especially grateful for the comments with regard to the validity of the asymptotic behavior and hope the derivation thereof should make for an insightful addition to the introductory many-body physics section.

Below, we address a few of the issues in detail:

Describing the slicing procedure (Fig.3) it would be of help for a reader to comment the method w.r.t the usual fixed size slicing (trotterization).

We don't really perform a "slicing procedure". We make it quite clear that this is a continuous time method. Discussing the absence of a systematic error from a non-existent discretization procedure seems rather pointless. Further, continuous time methods are quite prevalent these days, exactly for these advantages. We believe today Trotter-based methods are by no means the standard "gateway method" they used to be. Alluding to these methods this early in might give the reader the wrong idea. Thus, we respectfully disagree with the referee in this matter on pedagogical grounds.

Related to the discussion in sect. 3.4 the statistics of the diagram order should be improved according to my opinion. In both sections 2 and 3 a discussion on the sampling of the order of perturbation theory will be useful to the reader. In the datasets produced for the two level system (sect. 2) seems not to provide this information. Could the authors provide this information?

We believe we discuss the statistics of the diagram orders for the two-level system in Secs. 2.5 and 3.6. In the former, we give details on the percentages of the time spent in zeroth and higher orders; in the latter, Fig. 11 shows a histogram of the bare expansion orders.

In Eq. (25) but also in previous part of the paper the Weights, which are proportional to odd number of electron propagators are negative, perhaps a comment on this fact here or in the previous section should be needed.

We corrected this in the updated manuscript. The Green function and self-energy, and indeed all of the contributing diagrams, are strictly negative quantities which enables us to sample them without a sign problem by defining their MC weights with the opposite sign. This led us to "forget" about the physical minus sign in many places and indeed this minus sign does not appear in our code. However, for the benefit of readers unfamiliar with diagrammatic Monte Carlo, we agree that we should keep the notation in these notes consistent with the usual conventions found in many-body physics literature.

I also suggest the authors to supplement the paper with a list of exercises or propose exercises whenever possible. In the present of the paper on page 23,25 and 31 there are suggestions for possible exercises. I think it would be useful to regroup it on a devoted section or emphasize better in the text.

Perhaps a list of suggested exercises based on the codes (which are now proposed in the text) should be added at the end of the manuscript.

After careful consideration, we decided against amending the current manuscript with a dedicated section for exercises. The "exercises" mentioned in the text currently only concern technicalities and are mostly already "solved" in the code and would be of little pedagogical value to actually do. We considered developing a set of more meaningful exercises but see this outside the scope of this manuscript. In some sense, instructions to reproduce most of the figures are supplied as READMEs along with the code. We'd like to keep it that way, for ease of keeping them up-to-date with the corresponding source code. Smaller, pedagogical exercises for use in a workshop or school hands-on session would require additional effort to be of real use.

---

## Round 2 · Referee Report · Anonymous (Referee 2) · 2018-2-3

Strengths

1- These lecture notes provide a thorough introduction to Diagrammatic Monte Carlo for the Fröhlich polaron 2- Many important technical details, which are usually omitted, are explicitly given. 3- open source codes are made available

Weaknesses

1- The algorithm for the Fröhlich polaron dates back to 1998. More recent developments for many-fermion systems are only mentioned very briefly.

Report

The manuscript "Lecture notes on Diagrammatic Monte Carlo for the Fröhlich polaron" gives a detailed description of a Monte Carlo method that evaluates a series of Feynman diagrams for a specific model where no sign problem appears. Although the Diagrammatic Monte Carlo method was already described before in articles (the method was introduced in 1998!), the current lecture notes are still useful since they give much more details about the implementation and consider extensions such as sampling of the self-energy or bold diagrammatic Monte Carlo, which evaluates the series of diagrams built from dressed propagators. Moreover, links to sample codes are provided and more recent developments (such as the simulation of the Fermi polaron) are briefly discussed. I believe these lecture notes deserve to be published in SciPost Physics Lecture Notes since they will definitely be helpful to anyone who wants to learn this technique. They are clear and well-written. I do have a number of (minor) issues with the current manuscript which I would like to see resolved (some of these are just typos or errors, other remarks are given in order to render the manuscript more pedagogical). See the list of requested changes.

Requested changes

p. 4. "The main difficulty in the development of DiagMC is the sign problem,..." after which reference [11] is cited. However, this paper deals with the conventional sign problem when the CPU time scales exponentially with volume and inverse temperature. In the current case, the scaling is rather factorial with the order of expansion. I would reformulate and cite [11] in a more correct way.

p.8 section 2.3: The weight in the old configuration X is $W(X) = eˆ{-(\tau_2-\tau_1)hn_0}$. In fact, this is incorrect. Not the full weight is given, but only the factor which is altered. Though this might be obvious to anyone familiar with Monte Carlo, it might be confusing to someone who is learning the method.

p.8 section 2.4: I fail to see why the second estimator is not really better than the first one when done "on the fly" after each update. Could the authors provide some insight?

p. 10 The braket notation used in Eq. (10) is not explained. Please give more information about the used notation.

p. 11 The formula for the residue (Eq.(19)) is wrong. The correct formula should be 1/(1+X) with X the expression given in the r.h.s. of Eq.(19).

p. 12 "Our task consists of sampling over all possible diagrams for the bare Green function G, i.e…." Why bare? I believe you mean the dressed Green function.

p. 12 Eq (20) The integral over tau_4 should run from $\tau_3$ to $\tau$. Please correct this mistake.

p. 13 "The same can be done for other quantities of interest such as the bare Green function and the first order Green function." It is not clear to me what this sentence is trying to say. Please explain.

p. 14 Figure 7: the momentum $k'$ is wrong. It should be $k - q_1 - q_2$.

p. 15 section 3.4 "The necessary information needed to specify…" I do not understand this very long sentence. Why "and at least one of the phonon and electron momenta.." ? You already said you need to know the phonon momentum right before.

p. 21 Eq. (27) t should be tau.

p. 22. Why do you introduce formulas at finite temperature (Eqs. (31) and (32)). I think this might be confusing. It was mentioned earlier that calculations are done at zero temperature. Moreover, is $\beta$ the same as $\tau_{max}$? I understand you want to introduce an equidistant grid so that FFT can be used subsequently, but this does not require to introduce a finite temperature all of a sudden.

p. 23 $\omega_n = 2 \pi n \beta$. There is an obvious mistake here.

p. 25. I find the figure 16 quite confusing. There is E on the y-axis, on the x-axis and in the legend. I would like to see a more clear version of this figure.

p. 29 Eq (41) I don't see why the authors are using arrows. These are equalities, right?

p.29 Eq (42) and (43): the coupling has become $Vˆ2$, whereas $V$ was the volume before. The notation is not consistent with Eq (39).

p. 29 Eq (45) A limit is being calculated here. Why use $\approx$?

p. 31 section 5.3. Why is the diagram after INSERT unphysical? Do you not rather mean two-particle reducible?

p. 33 section 7: Please explain the sentence "In practice, bold DiagMC schemes rely on the $Gˆ2W$ scheme."

p. 35 section 8.3: "They can be tuned, even made infinitely strong". How can the interaction be infinitely strong? You probably refer to the fact that the scattering length can diverge. I suggest to write this instead.

Simple typos: bounary (p.7), consequenece (p.8), formula's (p. 12), wihtin (p.14)

  • validity: good
  • significance: good
  • originality: good
  • clarity: high
  • formatting: excellent
  • grammar: excellent

Author:  Jonas Greitemann  on 2018-03-05  [id 223]

(in reply to Report 2 on 2018-02-03)

We thank the referee for their time and effort to review our manuscript and are grateful to have received such a positive report. We appreciate the attention to detail that went into it and modified the manuscript to accommodate most of the criticism. Below, we examine some of their comments in detail:

p.8 section 2.4: I fail to see why the second estimator is not really better than the first one when done "on the fly" after each update. Could the authors provide some insight?

Updating the integrated quantity "on-the-fly" is an O(1) operation, but one that has to be done after every single update. Since the number of elementary updates to achieve an independent configuration scales at least with the simulation "volume" beta, updating the integral on-the-fly is not cheaper than calculating it explicitly upon measurement. All in all, the correlations in imaginary time are typically strong enough to make the second estimator converge only marginally faster than the first, and certainly not fast enough to make it worth the computational effort. This is true also of other MC methods with an efficient update scheme like SSE where one only considers the full "time" information in the measurements if the observable is "dynamic". In the latter case, one has to be very careful to avoid the measurement dominating the computational effort.

p. 22. Why do you introduce formulas at finite temperature (Eqs. (31) and (32)). I think this might be confusing. It was mentioned earlier that calculations are done at zero temperature. Moreover, is β the same as τmax? I understand you want to introduce an equidistant grid so that FFT can be used subsequently, but this does not require to introduce a finite temperature all of a sudden.

Indeed, beta == tau_max. We are at zero temperature, the beta is merely introduced to allude to the commonly used well-known definition of the Matsubara frequencies. We have added a remark to that effect.

p. 25. I find the figure 16 quite confusing. There is E on the y-axis, on the x-axis and in the legend. I would like to see a more clear version of this figure.

We hope to have cleared up any confusion regarding Fig. 16 by labeling the abscissa E' and the lines with E'=E and E'=Phi(E), respectively.

p. 33 section 7: Please explain the sentence "In practice, bold DiagMC schemes rely on the Gˆ2W scheme."

This sentence summarizes the paragraph in the language of Ref. 34, namely that bold DiagMC as it has been introduced in these notes is based on a resummation of the self-energy and in principle also of the irreducible polarization which corresponds to an expansion in z=iG^2W, W being the screened interaction. As stated just prior, the latter does not play any role in the polaron problem because the infinite bath is not renormalized by a single particle.

The following paragraph then discusses the possibility of basing bold DiagMC on the G^2W Gamma^2 expansion of Ref. 34, which is impractical because of the numerical problems in treating a 3-point vertex as opposed to 2-point self-energy and Green function.

---

## Round 3 · Author Response

Dear Editor,

we would like to thank you for the communicating the referee reports and are grateful for having received such positive and detailed responses. We have carefully considered the referees' comments and would hereby like to submit a revised version of our manuscript. Below, we give our views on some of the points raised in the reports:

First referee:

Describing the slicing procedure (Fig.3) it would be of help for a reader to comment the method w.r.t the usual fixed size slicing (trotterization).

We don't really perform a "slicing procedure". We make it quite clear that this is a continuous time method. Discussing the absence of a systematic error from a non-existent discretization procedure seems rather pointless. Further, continuous time methods are quite prevalent these days, exactly for these advantages. We believe today Trotter-based methods are by no means the standard "gateway method" they used to be. Alluding to these methods this early in might give the reader the wrong idea. Thus, we respectfully disagree with the referee in this matter on pedagogical grounds.

Related to the discussion in sect. 3.4 the statistics of the diagram order should be improved according to my opinion. In both sections 2 and 3 a discussion on the sampling of the order of perturbation theory will be useful to the reader. In the datasets produced for the two level system (sect. 2) seems not to provide this information. Could the authors provide this information?

We believe we discuss the statistics of the diagram orders for the two-level system in Secs. 2.5 and 3.6. In the former, we give details on the percentages of the time spent in zeroth and higher orders; in the latter, Fig. 11 shows a histogram of the bare expansion orders.

In Eq. (25) but also in previous part of the paper the Weights, which are proportional to odd number of electron propagators are negative, perhaps a comment on this fact here or in the previous section should be needed.

We corrected this in the updated manuscript. The Green function and self-energy, and indeed all of the contributing diagrams, are strictly negative quantities which enables us to sample them without a sign problem by defining their MC weights with the opposite sign. This led us to "forget" about the physical minus sign in many places and indeed this minus sign does not appear in our code. However, for the benefit of readers unfamiliar with diagrammatic Monte Carlo, we agree that we should keep the notation in these notes consistent with the usual conventions found in many-body physics literature.

I also suggest the authors to supplement the paper with a list of exercises or propose exercises whenever possible. In the present of the paper on page 23,25 and 31 there are suggestions for possible exercises. I think it would be useful to regroup it on a devoted section or emphasize better in the text.

Perhaps a list of suggested exercises based on the codes (which are now proposed in the text) should be added at the end of the manuscript.

After careful consideration, we decided against amending the current manuscript with a dedicated section for exercises. The "exercises" mentioned in the text currently only concern technicalities and are mostly already "solved" in the code and would be of little pedagogical value to actually do. We considered developing a set of more meaningful exercises but see this outside the scope of this manuscript. In some sense, instructions to reproduce most of the figures are supplied as READMEs along with the code. We'd like to keep it that way, for ease of keeping them up-to-date with the corresponding source code. Smaller, pedagogical exercises for use in a workshop or school hands-on session would require additional effort to be of real use.

Second referee:

p.8 section 2.4: I fail to see why the second estimator is not really better than the first one when done "on the fly" after each update. Could the authors provide some insight?

Updating the integrated quantity "on-the-fly" is an O(1) operation, but one that has to be done after every single update. Since the number of elementary updates to achieve an independent configuration scales at least with the simulation "volume" beta, updating the integral on-the-fly is not cheaper than calculating it explicitly upon measurement. All in all, the correlations in imaginary time are typically strong enough to make the second estimator converge only marginally faster than the first, and certainly not fast enough to make it worth the computational effort. This is true also of other MC methods with an efficient update scheme like SSE where one only considers the full "time" information in the measurements if the observable is "dynamic". In the latter case, one has to be very careful to avoid the measurement dominating the computational effort.

p. 22. Why do you introduce formulas at finite temperature (Eqs. (31) and (32)). I think this might be confusing. It was mentioned earlier that calculations are done at zero temperature. Moreover, is β the same as τmax? I understand you want to introduce an equidistant grid so that FFT can be used subsequently, but this does not require to introduce a finite temperature all of a sudden.

Indeed, beta == tau_max. We are at zero temperature, the beta is merely introduced to allude to the commonly used well-known definition of the Matsubara frequencies. We have added a remark to that effect.

p. 25. I find the figure 16 quite confusing. There is E on the y-axis, on the x-axis and in the legend. I would like to see a more clear version of this figure.

We hope to have cleared up any confusion regarding Fig. 16 by labeling the abscissa E' and the lines with E'=E and E'=Phi(E), respectively.

p. 33 section 7: Please explain the sentence "In practice, bold DiagMC schemes rely on the Gˆ2W scheme."

This sentence summarizes the paragraph in the language of Ref. 34, namely that bold DiagMC as it has been introduced in these notes is based on a resummation of the self-energy and in principle also of the irreducible polarization which corresponds to an expansion in z=iG^2W, W being the screened interaction. As stated just prior, the latter does not play any role in the polaron problem because the infinite bath is not renormalized by a single particle.

The following paragraph then discusses the possibility of basing bold DiagMC on the G^2W Gamma^2 expansion of Ref. 34, which is impractical because of the numerical problems in treating a 3-point vertex as opposed to 2-point self-energy and Green function.

---

## Round 3 · List of Changes

• In the abstract, more aspects and extensions that are addressed in these notes are advertised as such.
  • Sec. 1: more accurate citation of Ref. 11, as pointed out by referee #2
  • Sec. 2.3: Stressed that W(X) and W(Y) refer to the weights of only the segment of the diagram that is being altered in the update as pointed out by referee #2.
  • Sec. 3 (intro): Explained basis states for bra-ket notation and motivate path integral formalism, as requested by referee #2.
  • Sec. 3.1: Derivation of asymptotic form of the imaginary time Green function as requested by referee #1. The equations for E_k and Z_k are also derived and referred to by Eq. number. A mistake in the equation for Z_k, pointed out by referee #2, as been corrected. The limits of the approximation, implicit in the derivation, are now discussed.
  • Sec. 3.2: Clarified sentence concerning normalization of Green function and other quantities, as mentioned by referee #2.
  • Included minus sign in the Monte Carlo weight of the diagram and its constituents. As correctly pointed out by referee #1, the Green function (and self-energy) are strictly negative quantities.
  • Sec. 4.1: Added comment on the fictitious nature of beta in the Fourier transforms.
  • Sec. 5.1: Made the notation more precise: the self-energy is viewed as a functional of the Green function. The split into bare Green function and corrections (and their corresponding self-energy contributions) is more explicit now (written as a proper equality) and the primed self-energy (contribution due to delta G) is defined as such.
  • Sec. 5.3: Occurrences of "physical" / "unphysical" have been replaced with "two-particle irreducible" / "reducible" which is more precise and is what is meant in this context, as pointed out by referee #2.
  • Sec. 8.3: more precise statement on tunable scattering lengths in cold atom experiments
  • Various typos and other details
  • Chose thicker line width and more distinguishable line colors in most plots.
  • Relabeled the abscissa in Fig. 16 as E'.

---

## Editorial Decision

published